# Action history influences subsequent movement via two distinct processes

Welber Marinovic[1,2]*, Eugene Poh[2,3], Aymar de Rugy[2,4], Timothy J Carroll[2]*

[1]School of Psychology and Speech Pathology, Curtin University, Perth, Australia; [2]Centre for Sensorimotor Performance, School of Human Movement and Nutrition Sciences, The University of Queensland, Brisbane, Australia; [3]Department of Psychology, Princeton University, Princeton, United States; [4]Institut de Neurosciences Cognitives et Intégratives d'Aquitaine, CNRS UMR 5287, Université Bordeaux Segalen, Bordeaux, France

**Abstract** The characteristics of goal-directed actions tend to resemble those of previously executed actions, but it is unclear whether such effects depend strictly on action history, or also reflect context-dependent processes related to predictive motor planning. Here we manipulated the time available to initiate movements after a target was specified, and studied the effects of predictable movement sequences, to systematically dissociate effects of the most recently executed movement from the movement required next. We found that directional biases due to recent movement history strongly depend upon movement preparation time, suggesting an important contribution from predictive planning. However predictive biases co-exist with an independent source of bias that depends only on recent movement history. The results indicate that past experience influences movement execution through a combination of temporally-stable processes that are strictly use-dependent, and dynamically-evolving and context-dependent processes that reflect prediction of future actions.
DOI: https://doi.org/10.7554/eLife.26713.001

*For correspondence:
welber.marinovic@curtin.edu.au
(WM);
timothy.carroll@uq.edu.au (TJC)

Competing interests: The authors declare that no competing interests exist.

## Introduction

Animal survival depends upon the ability to execute movements that are customized to the current environmental context. Both general decisions about *what* to do, and the specifics of *how* actions should be executed, must take into account the identity, location and motion of physical objects in the animal's vicinity, as well as the current state of the animal itself. Multiple lines of ongoing research are devoted to revealing how the central nervous system meets this difficult challenge of linking multiple environmental and internal states with the generation of effective movements, but one established principle is that, in addition to the current context, each individual's *past actions* strongly influence action selection and execution. For example, parameters of reaching movements including initial direction, speed, and curvature, are biased to resemble the characteristics of recently executed movements (*Diedrichsen et al., 2010*; *Hammerbeck et al., 2014*; *Huang et al., 2011*; *Jax and Rosenbaum, 2007*; *Jax and Rosenbaum, 2009*; *van der Wel et al., 2007*; *Verstynen and Sabes, 2011*; *Chapman et al., 2010b*; *Chapman et al., 2010a*; *Wong and Haith, 2017*). Recent movement history also biases decisions about which action to perform when individuals are free to choose between multiple options (*He and Kowler, 1989*; *Hudson et al., 2007*), and affects the time taken to generate a response after it is specified from a range of alternatives (*Dorris and Munoz, 1998*; *Hyman, 1953*; *Hick, 1952*).

There are two general types of process by which movement history could affect subsequent motor behaviour. First, bias towards the characteristics of past actions could be driven by simple 'use-dependent' effects, in which the neural representations of repeated actions are increased. This

type of process could manifest on a short time-scale as a potentiation of synapses that are repeatedly activated (e.g. *Classen et al., 1998*; *Selvanayagam et al., 2016*; *Ziemann et al., 2004*), or in the longer term as a greater number of neurons tuned to a stimulus property or movement (*Chapman and Bonhoeffer, 1998*; *De Valois et al., 1982*; *Scott et al., 2001*), or a more tightly coupled network associated with a particular stimulus or response (e.g. *Wong et al., 2016*).

Alternatively, behaviour might be biased to resemble past actions due to a history-dependent *prediction* of actions likely to be required next. In this case, past experience would serve to prime the motor system to prepare, in advance of a final commitment to act, actions that are typically required in the relevant context. Behavioural biases would then emerge when an unexpected action is required at short notice, and movement is initiated before competition between the neural representations of potential actions is resolved. Indeed, there is converging evidence from studies of behaviour and neuronal recordings that primates represent multiple potential actions afforded by the sensory context in parallel (*Cisek and Kalaska, 2005*; *Gallivan et al., 2015*; *Gallivan et al., 2016*; *Klaes et al., 2011*; *Song and Nakayama, 2008*), and that decisions between these potential actions are reached through competitive interactions between sensory evidence and each individual's current internal neural state (*Afshar et al., 2011*; *Dorris et al., 2007*; *Forstmann et al., 2008*; *Pastor-Bernier and Cisek, 2011*; *Thura and Cisek, 2016*).

Because movement history provided the contextual information necessary to predict the probability of future action requirements in past experiments (e.g. *Wong and Haith, 2017*; *Verstynen and Sabes, 2011*; *Chapman et al., 2010b*; *Marinovic et al., 2017*), it is unclear to what extent movement direction biases are due to use-dependent processes that depend strictly on movement repetition, or due to history-dependent predictions of future action requirements. If both factors contribute, it is unknown how they interact, or are co-represented in the brain. Here we set out to dissociate these putative factors through a series of experiments involving control of movement preparation time, and sequences of two consecutive movements. We show that the effects of action history involve both dynamically-evolving processes reflecting prediction of future actions, and temporally-stable processes induced by movement repetition. Thus, past experience shapes future behaviour via multiple distinct mechanisms.

## Results

### Experiment 1 – Aiming bias is greater with reduced movement preparation time

We first sought to establish whether the effects of movement history are sensitive to the amount of time that people have available to prepare a response after a target is presented. To this end, we used the timed response paradigm to cue participants to initiate their movements in synchrony with a predictable signal (see *Figure 1B,C*). Participants made isometric wrist force pulses towards targets that were presented, in separate blocks, either 500 ms or 150 ms before the cue to initiate movement. Most movements were made to 'context targets' whose position was drawn randomly from a Gaussian distribution of mean 45° (SD = 7.5), but a subset of movements were made to 'probe targets' that were occasionally presented at one of five angular locations (see *Figure 1B*). If movements are biased to resemble frequently repeated actions because of a context-dependent prediction of future action requirements, then biases toward the centre of the target distribution should be greater when movements to probe targets were initiated following short than long preparation times (see e.g. *Marinovic et al., 2017*). This is because long preparation times should allow more time for neural activity to shift from an anticipatory state associated with preparation of more likely actions (i.e. to the centre of the context target distribution), to a state appropriate to initiate movement toward an unexpectedly presented probe target.

Our primary variable of interest was the angle between the initial direction of movement (i.e. 100 ms after movement onset) and a straight line to each probe target, hereafter referred to as directional error. *Figure 2A* shows the mean directional error for each probe position in both blocks of Experiment 1. The results for the short preparation time condition closely resemble those of *Verstynen and Sabes (2011)*, who used a reaction time task in which preparation time was not specifically manipulated, in that bias was greater for movements to probe targets further from the centre of the context target distribution. By contrast, bias was weak or absent for the long preparation

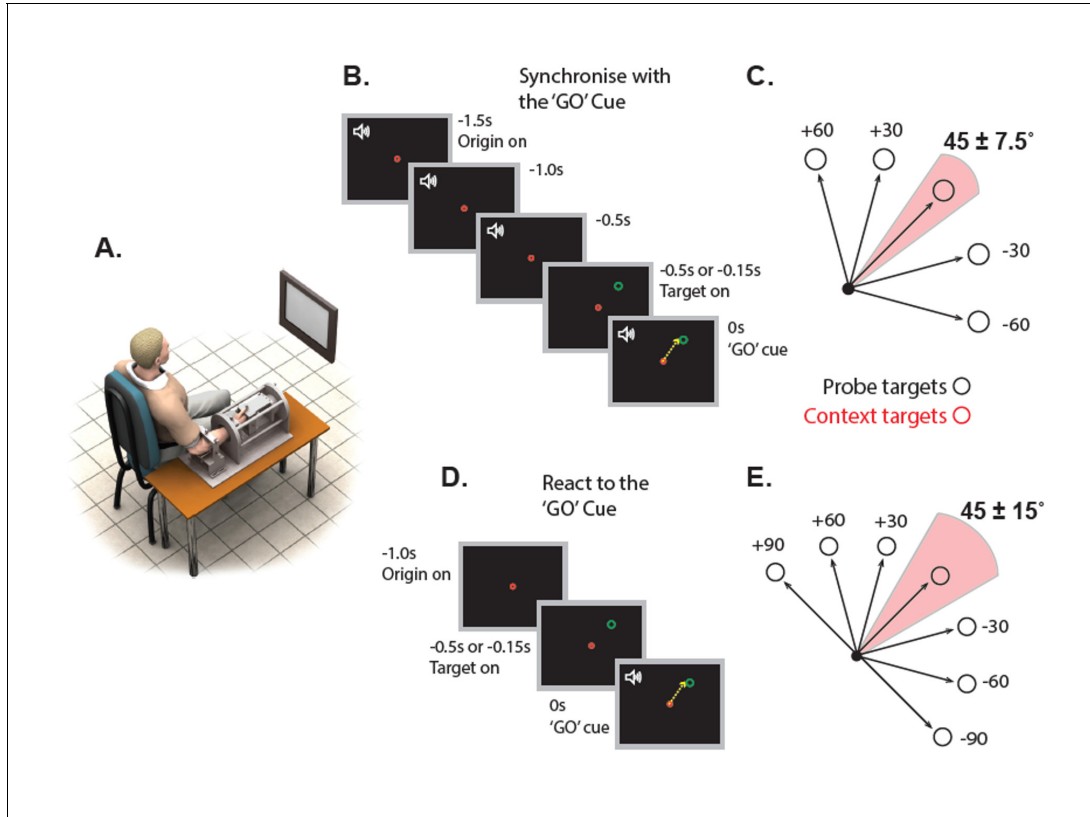

**Figure 1.** Experimental protocol and setup for experiments 1 and 2. (A) Illustration of the experimental configuration. (B) A schematic showing the Experiment 1 trial sequence using the timed response paradigm. Participants initiated their movements in synchrony with the final tone in a sequence of four. The probe target did not appear until either 500 (long preparation) or 150 ms (short preparation) before the fourth tone. (C) A schematic representation showing the locations of context (shaded pink area) and probe targets (grey) in Experiment 1. (D) Trial sequence for the reaction time task of Experiment 2. Participants initiated movements as soon as possible after an auditory 'GO' cue. The target location was presented either 150 ms (short preparation) or 500 ms (long preparation) prior to the 'GO' cue. (E) Same as C but for Experiment 2.
DOI: https://doi.org/10.7554/eLife.26713.002

condition, presumably because participants had sufficient time to fully respecify their intended action to accommodate the new target location prior to movement initiation. The two-way repeated measures (RM) ANOVA supported these conclusions. There were significant main effects of probe position, $F_{1.4, 12.8} = 23.01$, $p<0.0001$, and preparation time, $F_{1, 9} = 45.13$, $p<0.0001$, and a statistically significant interaction between probe position and preparation time, $F_{2, 18} = 39.02$, $p<0.0001$. Separate trend analyses for the short and long preparation blocks of trials showed a statistically reliable linear trend ($F_{1, 9} = 45.44$, $p<0.0001$) with a large slope for the short preparation block, slope = 0.56, 95% CI [0.41, 0.72], but no significant trend ($F_{1, 9} = 0.64$, $p=0.44$) and a small to negligible slope for the long preparation block, slope = 0.04, 95% CI [−0.04, 0.11]. The data indicate that preparation time plays a critical role in determining the magnitude of directional biases due to recent movement history, as would be expected if dynamic processes associated with prediction of future actions are involved.

*Figure 2B* shows the mean time available for movement preparation (the time from the presentation of the target until the initiation of the motor response) for each probe position in the short and long preparation blocks. As expected, participants were able to use the auditory cues to approximately match the required timings in each block (i.e. 150 ms for short preparation block, 500 ms for the long preparation). However, people had a tendency to initiate movement slightly before the GO cue in the long preparation condition (−72 ms), but slightly after the GO cue when preparation time was short (16 ms), as supported by a statistically reliable main effect of preparation time condition, $F_{1, 9} = 50.54$, $p<0.0001$, on time of movement initiation with respect to the GO cue. Early initiation of responses is typical in anticipatory timing tasks (*de Rugy et al., 2012b*; *Marinovic et al., 2009*),

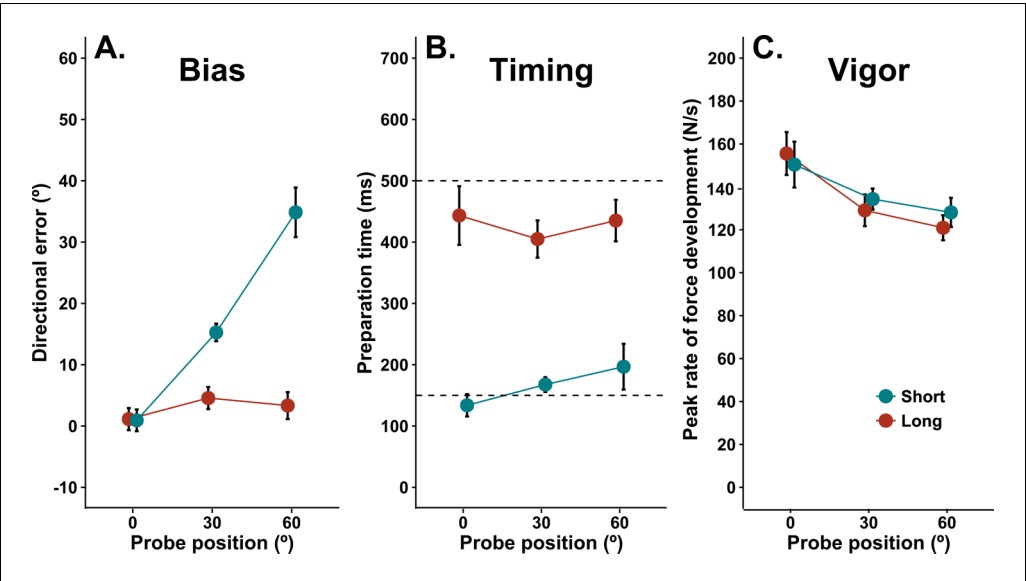

**Figure 2.** Effects of movement history in a timed response task. Effects of movement history on aiming bias (**A**), the time of movement initiation after target presentation (**B**), and movement vigor (**C**) for both long and short preparation time conditions in a timed response task. Plots show group mean values (±within subjects SE, see Materials and methods for details) of the median effect for each participant. Dashed lines in B indicate the time at which movement initiation was cued.

DOI: https://doi.org/10.7554/eLife.26713.003

The following source data is available for figure 2:

**Source data 1.** Source data for plots in panels 2a, 2b, 2c.

DOI: https://doi.org/10.7554/eLife.26713.004

so the relative delay in movement initiation for the short preparation time is consistent with a process that serves to oppose movement initiation when sensory information reflecting an unexpected goal is processed. However, the analysis of variance showed no statistically significant main effect of probe position, $F_{1.2, 10.8} = 0.47$, p=0.62, nor a significant interaction between preparation time and probe position, $F_{1.3, 11.9} = 1.93$, p=0.17.

The effect of movement preparation time on aiming basis that is apparent in *Figure 2* relies on median values from each participant. In *Figure 3a and b*, we show a more complete picture of the trial-by-trial inter-relationship between preparation time and target angle. Bias is plotted according to deciles ordered by movement preparation time. Here, each point plotted from top to bottom represents the average bias for trials initiated with the longest to shortest preparation times for each subject. For example, because there were 14 movements made to each target per condition, the bias value for each individual at the fifth percentile for preparation time is a weighted average of aiming biases from trials with the 14th and 13th shortest preparation times (i.e. the fifth earliest movement initiation time assuming 100 trials; actual values per condition obtained by linear interpolation within the 14 trials). *Figure 3a* shows that, for the long preparation time condition, bias appears relatively insensitive to trial-by-trial variations in movement preparation time. A RM ANOVA found no statistically reliable effects of probe position, $F_{2, 18} = 1.43$, p=0.26, nor (preparation time) deciles, $F_{9, 81} = 0.45$, p=0.89. The interaction between probe position and (preparation time) deciles was also not statistically significant, $F_{8.71, 78.4} = 1.01$, p=0.43. In contrast, as shown in *Figure 3b*, bias toward the central target increased as preparation time reduced for both peripheral targets (30° and 60°) under the time pressure of the short preparation condition. Here, the analysis of variance indicated significant main effects of probe position, $F_{1.33, 12.01} = 33.28$, p<0.0001, and (preparation time) deciles, $F_{9, 81} = 12.83$, p<0.0001. The interaction between probe position and (preparation time) deciles was also statistically significant, $F_{10.1, 91.6} = 4.81$, p<0.001. Follow-up polynomial contrast analyses showed reliable linear trends for bias to increase as preparation time reduced for probe targets at 30° (slope = −20.51, 95% CI [−26.92, −13.6], $F_{1, 9} = 31.59$, p<0.001) and 60°

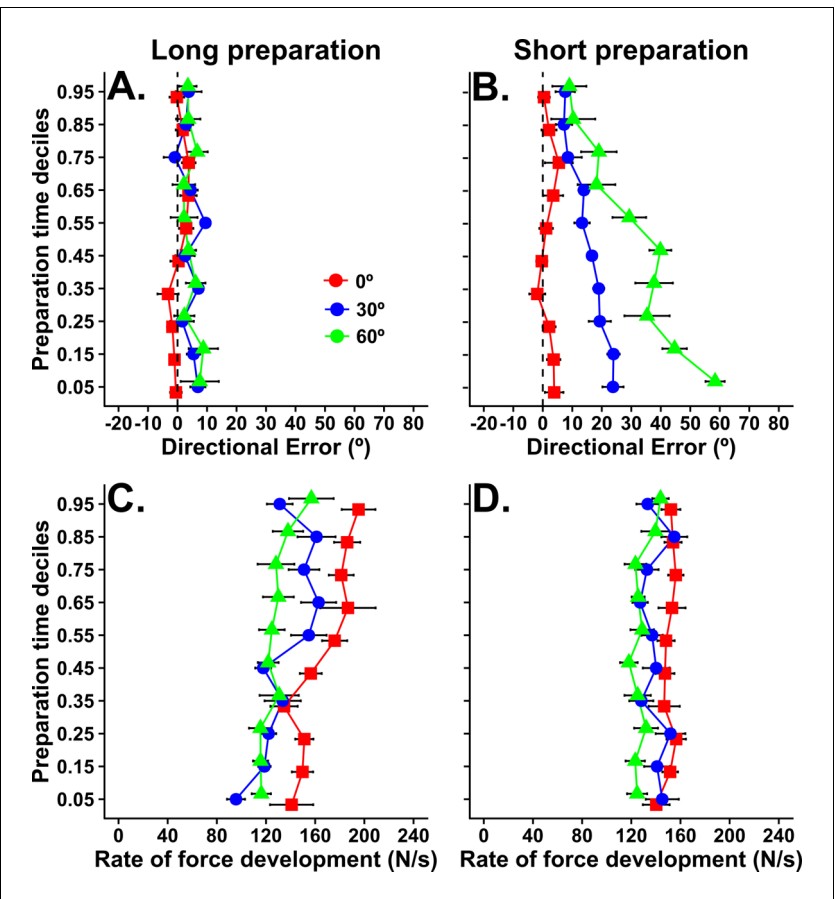

**Figure 3.** Plots showing how movement bias (top) and vigor (bottom) vary as a function of preparation time and target angle within each preparation time condition in experiment 1. Group average (and within-subjects SE) values for bias and vigor are plotted for trials corresponding to each preparation time decile. That is, a value at the fifth percentile for preparation time is the bias or vigor measured on the trial in which the available preparation time was at the fifth percentile (i.e. fifth shortest preparation time assuming 100 trials).

DOI: https://doi.org/10.7554/eLife.26713.005
The following source data is available for figure 3:

**Source data 1.** Source data for plots in panels 3a, 3b, 3c, 3d.
DOI: https://doi.org/10.7554/eLife.26713.006

(slope = −50.51, 95% CI [−63.15, −38.57], $F_{1,\,9}$ = 58. 1, p<0.001), but not in the direction of the average distribution of targets (slope = −0.62, 95% CI [−6.22, 6.4], $F_{1,\,9}$ = 0.18, p=0.89). The non-zero slopes demonstrate that directional biases are largest for the shortest preparation times, and progressively decrease as a function of the time available to prepare movement on any given trial *within* the short preparation block. The data also exclude the possibility that the median effects were due to a bi-modal relationship, in which very early movement initiations (i.e. guesses) were directed towards the expected target, whereas late movement initiations were directed accurately toward the peripheral probe targets. Note that the confidence intervals for the slopes at different targets do not overlap, implying that bias increased more, as a function of each individual's observed range of preparation times, as the distance between the probe target and the centre of the target distribution increased. Although this form of analysis does not inform about the absolute rate at which bias dissipates with additional preparation time, it provides strong evidence that bias had consistent temporal dependency across subjects for all peripheral targets.

Because response vigor can co-vary with reaction time for tasks requiring rapid eye movements (*Takikawa et al., 2002b*; *Itoh et al., 2003*), we also analysed the vigor of movements made to each probe target, defined as the peak of rate force development. The analysis (*Figure 2C*) showed that

movement vigor decreased as probe target angle departed from the repeated direction for both preparation time conditions ($F_{1.7, 15.2} = 11.8$, $p<0.001$), but that there was no significant effect of movement preparation time condition ($F_{1,9} = 0.05$, $p=0.83$) nor an interaction between these factors ($F_{1.2, 10.9} = 0.98$, $p=0.36$). These results suggest that recently repeated actions are executed more vigorously than actions that have been executed less frequently. However, in stark contrast to the results for directional biases, preparation time had little impact on the vigor of response execution when people could precisely anticipate the time of movement initiation. This dissociation is further illustrated by the plots of changes in vigor according to preparation time deciles shown in *Figure 3C,B*. Vigor was similar irrespective of movement preparation time in the short preparation time condition for all three targets (main effect of target: $F_{2, 18} = 4.94$, $p=0.019$; main effect of deciles: $F_{9, 81} = 1.08$, $p=0.38$; interaction: $F_{8.9, 80.1} = 0.64$, $p=0.75$), and tended to increase as movement initiation was delayed in the long preparation time condition (main effect of target: $F_{1.2, 10.8} = 15.07$, $p=0.002$; main effect of deciles: $F_{6.1, 55.3} = 5.52$, $p<0.001$; interaction: $F_{9.55, 85.6} = 1.18$, $p=0.31$). Consistent with the main effect of deciles in the long preparation condition, follow-up trend analyses indicated significant linear trends for increasing vigor as preparation time increased for movements to the more central probe targets (0° probe target: slope = 63.06, 95% CI [31.9, 103.5], $F_{1, 9} = 11.39$, $p=0.008$; 30° probe target: slope = 53.00, 95% CI [32.99, 79.16], $F_{1, 9} = 17.38$, $p=0.002$), but relatively smaller for the probe target at 60° (slope = 34.67, 95% CI [7.99, 65.5], $F_{1, 9} = 5.06$, $p=0.051$). Note that the 95% confidence intervals of these slopes overlap for the three targets, and the mean values are small with respect to the confidence intervals, indicating that the trend to increased vigor as movement initiation was delayed was weak across subjects. Nonetheless, it seems clear that the effect of action history on response vigor is distinct from the time-dependent effects on movement bias shown in 3A and B. The data suggest a dissociation between the neural processes that lead to biases in different parameters of the movement (i.e. spatial metrics versus vigor).

## Experiment 2 – Bias depends on the interaction between preparation time and the urgency to move

In Experiment 1, we used the timed response paradigm to control the time at which participants initiated movement, and found that movement biases were larger when preparation time was short. In Experiment 2, we examined the effects of preparation time using a reaction time task, since this paradigm informs whether response time benefits previously reported for repeated actions (e.g. *Dorris and Munoz, 1998*) depend on available preparation time. The paradigm also more closely resembles previous studies on history-dependent aiming effects (e.g. *Verstynen and Sabes, 2011*). In this case, although there was no explicit deadline for movement initiation, feedback of reaction times after each trial was used to motivate fast responses to the imperative cue. In separate blocks, the target was presented either 150 ms or 500 ms prior to an auditory 'GO' signal. The subjects were instructed to initiate their movements as fast as possible after they heard the GO signal. The basic task parameters were otherwise similar to those of experiment 1, except that we included an additional set of probe targets at 90° either side of the target distribution centre to more fully characterise the spatial tuning of any bias effects, and increased the width of the Gaussian distribution of target locations (mean = 45°; SD = 15°) from which context trials were randomly drawn.

*Figure 4A* shows the directional errors in the long and short preparation blocks across all four (±90°) probe positions. The pattern of results appears qualitatively similar to those obtained in Experiment 1, such that bias was larger for probe targets further from the centre of the context target distribution for the short but not the long preparation block. The analysis of variance supports this impression, because there were main effects of probe position, $F_{3, 27} = 4.37$, $p=0.012$ and preparation time, $F_{1, 9} = 5.35$, $p=0.046$, and an interaction between probe position and preparation time, $F_{3, 27} = 3.51$, $p=0.028$. As per Experiment 1, a trend analysis revealed a statistically significant linear trend with a positive slope for the short preparation block, slope = 0.15, 95% CI [0.047, 0.26], $F_{1, 9} = 6.09$, $p=0.036$, but a non-significant linear trend with a relatively smaller slope for the long preparation block, slope = 0.016, 95% CI [−0.014, 0.05], $F_{1, 9} = 0.96$, $p=0.35$. Note that these biases in movement direction are much smaller than those observed in experiment 1. This probably relates to the fact that the overall preparation times were much larger in experiment 2, due to the reaction time task paradigm, and to the differences in the width of the context target distribution used in the two studies (*Verstynen and Sabes, 2011*).

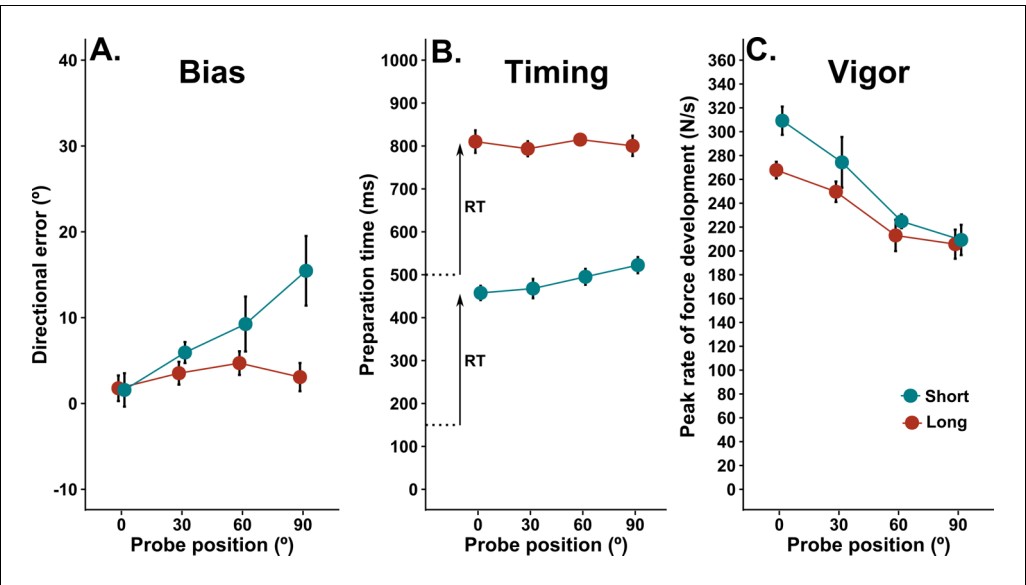

**Figure 4.** Effects of movement history in a reaction time task. Effects of movement history on aiming bias (**A**), the time of movement initiation after target presentation (**B**), and movement vigor (**C**) for both long and short preparation time conditions in a reaction time task. Plots show group mean values (±within subjects SE) of the median effect for each participant. Short, dashed lines in B indicate the time of the GO cue to which subjects had to react in each condition. The reaction time (RT) from the GO cue to movement initiation is indicated by the arrowhead lines.

DOI: https://doi.org/10.7554/eLife.26713.007

The following source data is available for figure 4:

**Source data 1.** Source data for plots in panels 4a, 4b, 4c.

DOI: https://doi.org/10.7554/eLife.26713.008

*Figure 4B* shows the mean preparation time available from target presentation until movement initiation for each probe location in the long and short preparation blocks. As expected, the overall preparation time was much greater for the long than the short preparation time condition, but it is of particular interest to examine the effect of movement history on the reaction time from the GO signal to movement initiation (see arrowhead lines in *Figure 4B*). Previous work showed that saccadic reaction times are typically shorter for eye movements toward targets that are more frequently presented (*Dorris and Munoz, 1998*). Here we found a similar effect for the short preparation condition, but not the long preparation condition. The analysis of variance showed main effects of probe position, $F_{3, 27} = 4.73$, p=0.009, and preparation time condition, $F_{1, 9} = 93.78$, p<0.0001. The interaction between preparation time and probe position was also statistically significant, $F_{3, 27} = 4.04$, p=0.017. Polynomial trend analysis showed a statistically significant linear trend for the short preparation block, slope = 0.74, 95% CI [0.42, 1.03], $F_{1, 9} = 20.7$, p=0.001, but not for the long preparation block, slope = −0.03, 95% CI [−0.45, 0.42], $F_{1, 9} = 0.01$, p=0.91. Although the slope confidence interval includes zero for the long preparation block, and not the short preparation block, the intervals are wide with respect to the mean effect in both cases, illustrating considerable inter-subject variability. However, the overall pattern of results illustrates that there was a time cost for the initiation of movement as the angle between the probe target and the centre of context target distribution increased when preparation time was short, but that any effect of target location on reaction time was weaker and more variable when preparation time was long.

*Figure 5A* shows that, as was the case in the timed response task of experiment 1, directional bias was relatively insensitive to trial-by-trial variations in movement preparation time in the long preparation reaction time condition. The RM analysis of variance indicated a lack of statistically significant effects of probe position, $F_{3, 27} = 1.99$, p=0.14, and preparation time deciles, $F_{9, 81} = 0.81$, p=0.61. Similarly, the interaction between probe position and preparation time deciles was not statistically significant, $F_{13.51, 121.6} = 1.10$, p=0.36. In contrast, bias toward the central target increased

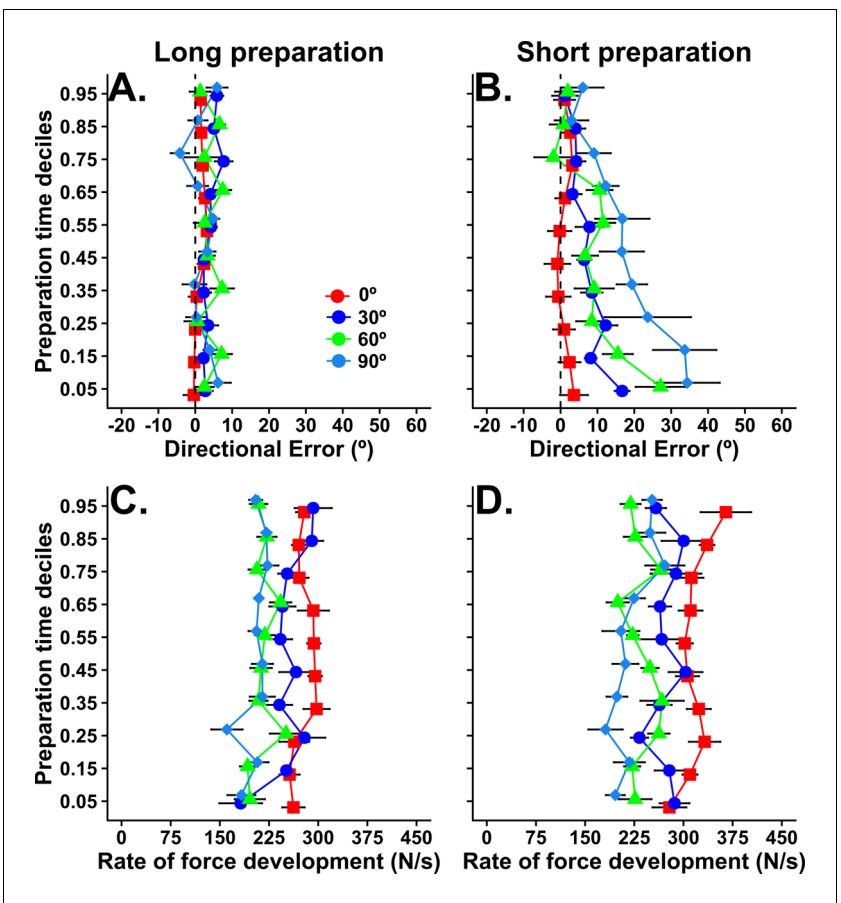

**Figure 5.** Plots showing how movement bias (top) and vigor (bottom) vary as a function of preparation time and target angle within each preparation time condition in experiment 2 (reaction time task). Group average (and within-subjects SE) values for bias and vigor are plotted for trials corresponding to each preparation time decile (as explained in *Figure 3*).

DOI: https://doi.org/10.7554/eLife.26713.009

The following source data is available for figure 5:

**Source data 1.** Source data for plots in panels 5a, 5b, 5c, 5d.
DOI: https://doi.org/10.7554/eLife.26713.010

as preparation time reduced for all peripheral targets (30°, 60° and 90°) when preparation time was short. As in Experiment 1, the RM ANOVA found statistically significant effects of probe position, $F_{1.55, 14.02} = 6.18$, p=0.002, and preparation time deciles, $F_{2.84, 25.6} = 5.01$, p=0.008. However, the interaction between probe position and preparation time deciles was not statistically significant, $F_{7.18, 64.7} = 1.86$, p=0.089. Overall, these results are qualitatively consistent with those from experiment 1 in that there is a tight, trial-by-trial coupling between directional bias and the amount of time that elapses between the target presentation and movement initiation, but that this effect only occurs under time pressure.

The results of experiment 2 emphasize the point that the amount of time available between target presentation and movement onset is a critical factor that determines the extent to which initial movement direction is biased according to movement history. However, a comparison between the preparation times available in experiments 1 and 2 reveals an interesting paradox. The time between target presentation and movement initiation was similar (at around 450–500 ms) for the long preparation condition of experiment 1 (see red dots in *Figure 2B*) and the short preparation time of experiment 2 (see blue dots in *Figure 4B*), and yet there was a clear discrepancy in the degree of aiming bias between these conditions. When the required time of movement initiation was uncertain in the reaction time task, 500 ms appeared insufficient to overcome a tendency to aim toward the

most likely next target. However, when the timed response protocol made the required time of movement execution predictable in experiment 1, participants were able to accurately aim to peripheral context targets within 500 ms. The data indicate that the amount of time available to process target location information prior to movement initiation is not the only factor that determines the behavioural effects of movement history. Rather, it appears that the urgency of response requirements interacts with preparation time. We return to this issue in the discussion, because it has bearing on the likely neural implementation of history-dependent biases.

The analysis of peak rate of force development again showed a general trend for movement vigor to decrease with increasing probe target angles from the repeated direction (main effect for target angle; $F_{1.6, 14.8} = 11.1$, p=0.002). However, in contrast to experiment 1, vigor was also greater for movements made in the short than the long preparation condition (main effect of condition; $F_{1, 9} = 5.99$, p=0.036). The interaction between preparation time condition and target angle was not statistically significant ($F_{2.5, 23} = 2.4$, p=0.1). The plots of changes in vigor according to preparation time deciles in *Figure 5C,D* support these analyses. For both long preparation and short preparation time conditions, the RM ANOVAs indicated statistically significant effects only for probe position (Long preparation: $F_{1.34, 12.07} = 9.11$, p=0.007; Short preparation: $F_{1.82, 16.4} = 14.02$, p<0.001). These results indicate that vigor was greater for movements towards central targets, but relatively independent of preparation time.

It is also of note that the grand average, peak rate of force development in the reaction time condition (244 N/s) was almost double that observed in the timed response condition of experiment 1 (136 N/s). Taken together, the results of both experiments suggest that movement vigor is reduced for actions that have been rarely executed in the recent past, irrespective of time constraints. This effect appears superimposed upon a more general effect associated with the task conditions, which may reflect the predictability of *when* an action must be initiated. For example, vigor was equivalent irrespective of available preparation time in experiment 1, when explicit cues were provided to facilitate precise anticipation of the required movement initiation time in both conditions. Moreover, vigor was much higher overall when movement initiation time was less certain in experiment 2 (see also *Mattes and Ulrich, 1997*), and highest in the short preparation condition which provided the least advanced information regarding the timing of the GO signal of all conditions (i.e. the only cue was the target appearance at 150 ms prior to the GO signal, compared with target appearance at 500 ms prior to the GO signal in the long preparation condition).

## Experiment 3 – Bias varies as a function of angle from a repeated action in the absence of target uncertainty

The results of experiments 1 and 2 show that recent movement history can lead to substantial aiming biases when a movement must be generated to an unexpected target location at short notice. In contrast, bias was weak or absent when participants were informed that movement to a rarely-visited target would be required 500 ms into the future. This pattern of findings suggests that bias under the conditions of these experiments was primarily due to a time-sensitive process that reflects advanced preparation of actions that are more likely to be required next, rather than a use-dependent process that is strictly dependent on recent movement history. However, previous work suggests that movement repetition can induce strictly use-dependent effects in some cases. For example, involuntary movements evoked by transcranial magnetic stimulation (TMS) of the motor cortex can be biased towards the direction of a repeated voluntary movement (*Classen et al., 1998*; *Selvanayagam et al., 2011*), and small biases occur towards the direction of previous movements, rather than future movements, when participants perform movements to a predictable sequence of targets with monotonically changing angles (*Verstynen and Sabes, 2011*). We therefore sought evidence for the existence of 'pure' use-dependent bias, using sequences of two consecutive movements that eliminated target location uncertainty.

Here, participants completed two blocks of trials, in which bias was measured for movements to a single probe target (first movement step) as a function of the direction of a second movement made to a series of 'fixed' targets (*Figure 6A,B*). One block was performed with the probe target at 90, and the other was performed with the probe target at 22. The order in which these were performed varied randomly for different subjects. Each fixed target was presented for 11 consecutive trials, but the order of fixed target presentation within a block differed randomly across subjects. This design removed all target-location uncertainty, and allowed us to plot the full tuning function of

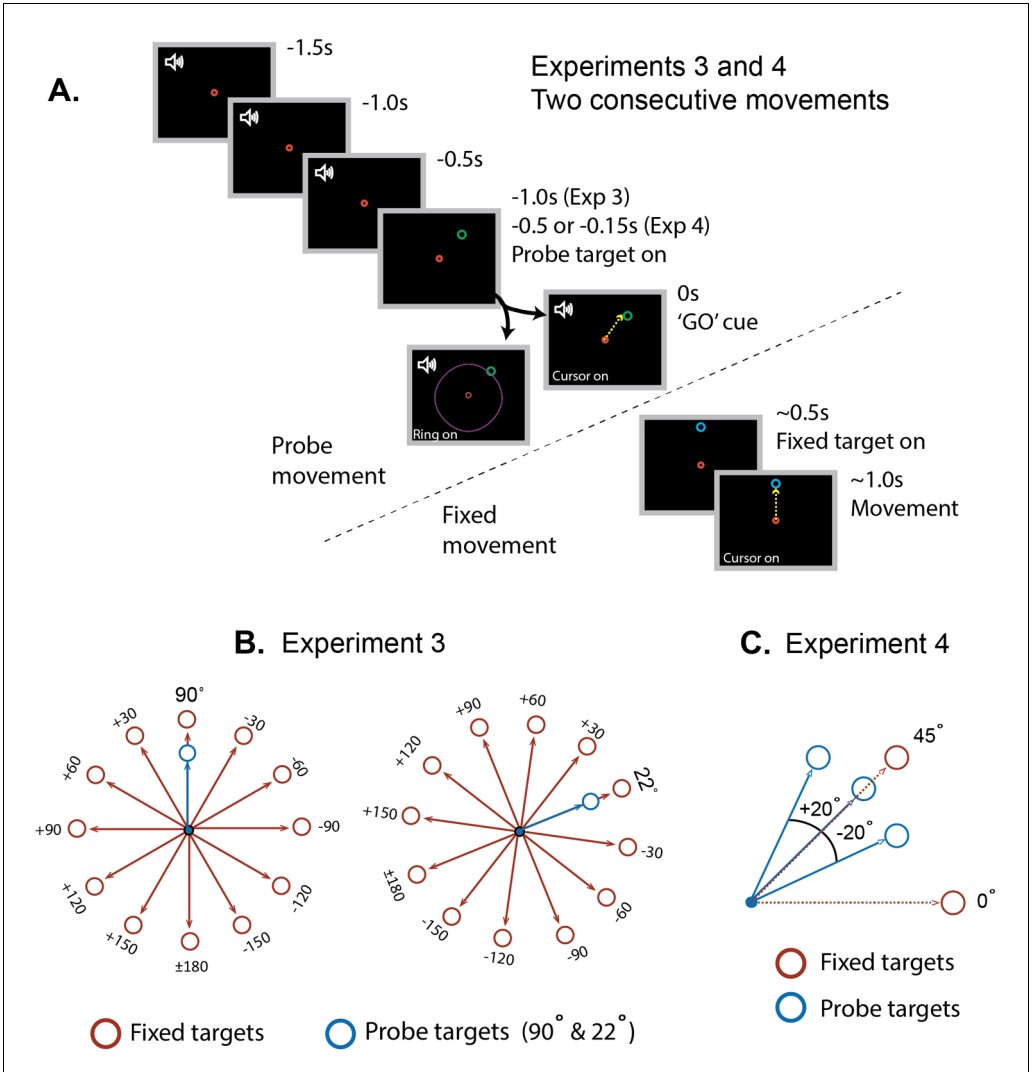

**Figure 6.** Experimental protocol and setup for experiments 3 and 4. (**A**) A schematic of a trial comprising a sequence of two consecutive movements using the timed response paradigm. Participants initiated their movements in synchrony with the final tone in a sequence of four. The probe target did not appear until 1000 ms (Experiment 3), 500 ms (Experiment 4, long preparation) or 150 ms (Experiment 4, short preparation) before the fourth tone. After participants acquired the probe target and returned the cursor to the origin, the fixed target was presented, signalling that the second movement should be made immediately. (**B**) Schematic representation of Experiment 3. The context targets were placed either at 22° (left) or 90° (right). Fixed targets were positioned at 30° intervals throughout a full 360° range around the context targets (30° steps) and participants performed movement sequences to pairs of targets in blocks of 11. (**C**) Schematic representation of Experiment 4. The probe target appeared at 45° more often (60% of the trials) than the two flanker locations (20% each). The fixed targets were positioned at 0° and 45° in separate blocks, and required 125% of the force required to reach the probe targets.

DOI: https://doi.org/10.7554/eLife.26713.011

any 'pure' history-dependent bias effect. Critically, we also removed visual feedback of movements made to probe targets. We suspected that a failure to detect substantial bias effects due to strictly use-dependent processes in the first two experiments occurred because movement errors due to bias were observable and therefore may have been corrected. Thus, error-based learning may have masked strictly use-dependent bias effects in these circumstances. We therefore anticipated that removing visual feedback during assessment of bias should provide the optimal conditions to study the properties of use-dependent bias.

*Figure 7A* shows the average directional biases (collapsed across the 22° and 90° probes) as a function of the relative angle between probe and fixed-targets. Note that the directional error here is the difference between the angle of force exerted when both movements in the double step sequence were made towards the probe target (baseline angle) and the angle of force exerted when moving between the probe target and fixed targets located from 30° to 180° away. Note also that, because the entire tuning function was derived from movements to the same two targets, inherent biomechanical or perceptual biases associated with this direction cannot influence the tuning functions. The analysis of variance showed a significant effect of fixed target position, $F_{3.75, 51.4} = 10.68$, $p<0.001$. Polynomial trend analysis showed that the linear ($F_{1, 17} = 9.15$, $p=0.008$) and quadratic ($F_{1, 17} = 40.88$, $p<0.001$) trends were statistically significant. Simple one-sample t-tests of the errors against 0 were statistically significant for fixed-targets at 30°, 60°, 90° and 120° (30°: 95% CI [4.0, 7.7]; 60°: 95% CI [4.52, 8.5]; 90°: 95% CI [3.29, 5.98]; 120°: 95% CI [2.51, 7.38]), but not for other targets.

An important issue that was not the specific focus of the current study is the temporal dynamics according to which bias effects accumulate over multiple trials. Since different numbers of movements to probe and context trials were performed in our different experiments, this issue is also relevant for comparisons of bias results between our experiments. To address this issue, we compared the median bias from the first two movements made during trials involving each fixed target with those from the last two movements. As shown in *Figure 7A,* a comparison between the median values obtained in early and late trials suggests that the errors tended to be larger as additional movements were executed. This effect was more pronounced for fixed targets at 30 (95% CI for difference between early and late trials [−4.40, 0.82]) and 60 (95% CI [−5.9,−0.5]) than for targets at 90 (95% CI [−4.68, 0.74]), 120 (95% CI [−4.72, 1.95]), 150 (95% CI [−3.96, 2.18]) and 180° (95% CI

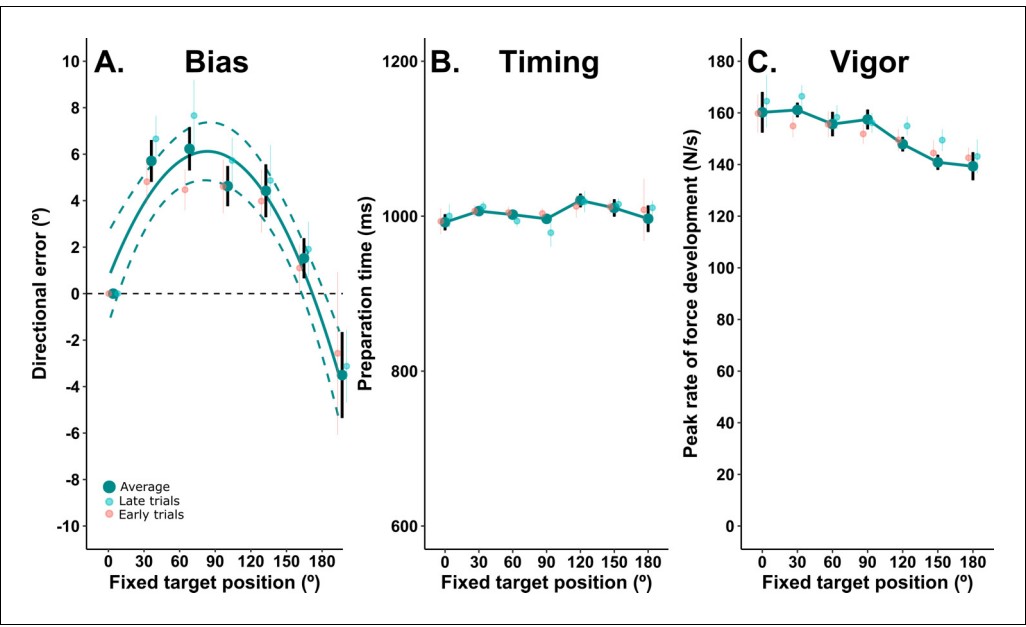

**Figure 7.** Movement bias as a function of angle from repeated target. (A) Group mean baseline-subtracted biases (±within subjects SE) as a function of the angular separation between targets. A second order polynomial fit (±95% CI) to the bias is shown to quantify the parameters of the tuning function (adjusted $R^2 = 0.90$). The time of movement initiation with respect to target presentation (initiation was cued at 1 s) is shown in **B**, and the vigor of movement as a function of the angular separation between targets is shown in **C**. The smaller circles in all plots show the median values for the first and last two trials that comprise each grand average.

DOI: https://doi.org/10.7554/eLife.26713.012

The following source data is available for figure 7:

**Source data 1.** Source data for plots in panels 7a, 7b, 7c.
DOI: https://doi.org/10.7554/eLife.26713.013

[−9.46, 6.4]). In sum, this analysis suggests that the effects of use-dependent biases can summate over repeated trials.

There was no main effect of target position for preparation time (*Figure 7B*, $F_{2.37, 40.4}$ = 2.01, p=0.14), but the main effect for target position was statistically significant for vigor (*Figure 7C*, $F_{3.29, 55.9}$= 3.79, p=0.013). This effect on vigour was associated with a significant linear trend, ($F_{1, 17}$ = 8.15, p=0.011, slope = −0.13, 95% CI [−0.22,–0.05], suggesting that the tendency, observed in the presence of target uncertainty, for movements to be more vigorous when their direction approaches that of a repeated action, persists when all target uncertainty is removed. As shown in *Figure 7C*, any cumulative effect of the number of trials performed on movement vigor was small, and all 95% confidence intervals of the difference between early and late trials included 0. This reinforces the point that movement vigor is susceptible to use-dependent effects of movement history, in the absence of time-dependent, predictive processes.

When comparing bias effects between experiments, it appears that the 'pure' repetition-dependent bias identified in experiment 3 is weaker (i.e. <7° vs >15°) and more local than the time-sensitive effects exposed experiments 1 and 2. Even more strikingly, there is an apparent absence of strictly use-dependent bias effects in experiments 1 and 2, despite clear evidence of such in experiment 3. This may relate to the fact that full visual feedback of movement trajectories was available to subjects in the first two experiments. We speculate that the processes that cause use-dependent biases are a general consequence of repeated action, but that the behavioural expression of such biases can be masked by error-based learning. Importantly, the bias distribution in *Figure 7A* peaks at 77°, according to a quadratic polynomial fit (adjusted $R^2$ = 0.90). This corresponds to a monophasic pattern of bias, peaking around 50–80°, that we recently observed when bias was probed with equally likely targets from 30 to 90 s following a bout of repeated movements to a single direction. In that paper, the data were well-fit by simulated activity-dependent weight-changes in a simple network comprising cosine-tuned units (*Selvanayagam et al., 2016*). Although extremely simple, the simulation illustrates how an increase in the relative contribution of a subset of directionally-tuned units within a neuronal population inevitably leads to local bias effects. In contrast, bias increased monotonically to 90° when target location was uncertain in experiment 2 in the current study, and in the study by *Verstynen and Sabes (2011)*. The discrepancies in bias tuning functions between conditions with and without target uncertainty suggest differences in the neural processes that underlie repetition-dependent versus action prediction biases. In experiment 4, we explore whether these processes can be experimentally dissociated, and if so, how they interact.

## Experiment 4 – Biases due to use-dependent and action prediction processes are experimentally separable

In the final experiment, we studied the interaction of biases due to the action prediction versus use-dependent effects of recent movement history. We again asked participants to perform sequences of two consecutive movements: the first movement was to one of the three context targets, and the second movement was to a fixed target that either coincided with, or was displaced from, the centre of the context target distribution (see *Figure 6C*). The three probe targets were presented with unequal probability; the central target at 45° was presented on 60% of trials, whereas each flanker target was presented on 20% of trials. The fixed targets were positioned at 0° and 45° in separate blocks, and required 25% more force to acquire than probe targets. *Figure 8* shows aiming errors for movements made towards the three probe targets. Note that, in all conditions, any advanced preparation of the action most likely to be required next should bias movements toward the central context target at 45° (see blue arrows in inset schematic plots). By contrast, movements to the fixed target provided no information about the probability of the next required action, so any differences in bias between blocks involving the different fixed targets should reflect pure use-dependent processes (see red arrows in *Figure 8*).

*Figure 8A* shows the average directional errors that participants made under long and short preparation conditions when the fixed target coincided with the central probe target (45°), and *Figure 8B* shows the same effect when the fixed target was displaced from the central probe target (i.e. at 0°). The pattern of errors appears very similar for the two fixed target conditions, except that all movements seem uniformly displaced towards 0° when the fixed target was located at 0° (i.e. resulting in more negative errors). This impression was supported by a three-way RM analysis of variance (preparation time [2] x fixed target [2] x probe target [3]), which showed a statistically significant

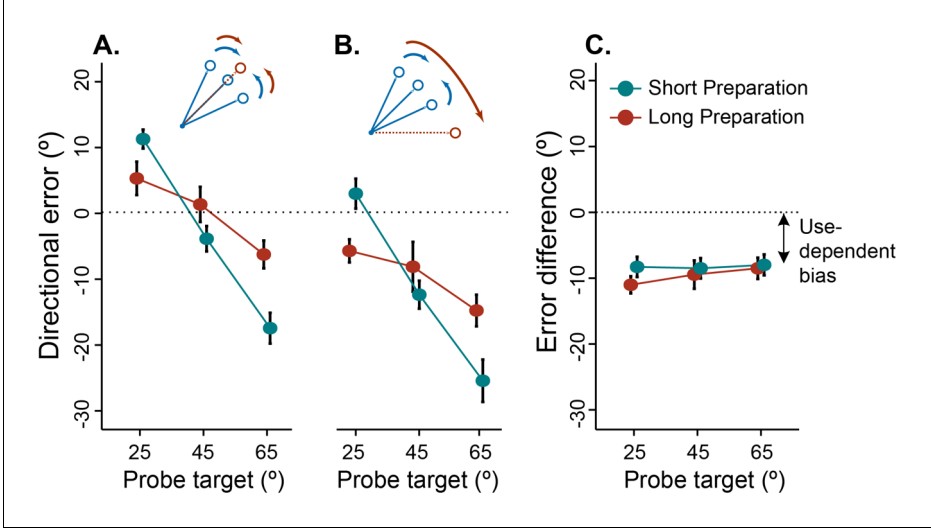

**Figure 8.** Dissociation between use-dependent and action prediction biases. (A, B) Group average (±within subjects SE) angular errors from each probe target for the two preparation time conditions and fixed targets. Counter-clockwise errors are depicted as positive, such that the pattern of errors in A represents biases toward the centre of the probe target distribution. The inset schematic plots illustrate the locations of the probe (blue) and fixed (red) targets, and the expected bias effects due to pure 'use-dependent' effects (red arrows) and 'action prediction' effects (blue arrows). Error distributions were similar for the two fixed target conditions (A and B), but were offset towards the fixed target when it was located at 0° (B). As in experiments 1 and 2, bias was greater for short than long preparation time for both fixed targets. However, the differences in errors between the conditions for which the fixed target was at 0° and for which the fixed target was at 45° (C), were similar for all probe targets and both preparation times. This error difference reveals that pure 'use-dependent' bias effects of recent movement history are insensitive to movement preparation time.

DOI: https://doi.org/10.7554/eLife.26713.014

The following source data is available for figure 8:

**Source data 1.** Source data for plots in panels 8a, 8b, 8c.
DOI: https://doi.org/10.7554/eLife.26713.015

main effect for the position of the fixed target, $F_{1, 13} = 28.42$, p<0.001, but no interaction effects involving this factor (all p>0.31). There was a statistically significant main effect of probe target position, $F_{1.3, 17.29} = 48.92$, p<0.0001, which illustrates that for the condition where the fixed target coincided with the central probe target (*Figure 8A*), movements toward the flanker targets tended to be biased toward the central target. As found in experiments 1 and 2, aiming bias was greater when there was less time to prepare a movement between target presentation and the GO signal, as supported by a significant interaction between preparation time condition and probe position, $F_{2, 26} = 29.63$, p<0.0001.

*Figure 8C* shows the differences in aiming errors for each corresponding target and preparation time condition between the two fixed target conditions. Remarkably, there were no statistically significant differences as a function of probe target position (main effect of probe target position: $F_{2, 26} = 0.27$, p=0.76), movement preparation time (main effect of preparation time: $F_{1, 13} = 1.09$, p=0.31) nor an interaction between these factors (Interaction between probe target position and preparation time: $F_{2, 26} = 0.38$, p=0.68). For equivalent probe target and preparation time conditions, all movements were biased towards the fixed target at 0° by a similar amount, with means ranging from −7.9° to −11° and overlapping confidence intervals; (all means were statistically different from a reference value of zero, t-tests: all p<0.007; Upper 95% CI ranging from −3.5 to −7.9; Lower 95% CI ranging from −11.8 to −17.7). These biases are larger than those observed in Experiment 3 (peaking at 77°) and may reflect a cumulative effect associated with the larger number of trials in Experiment 4. This shows that the final movement direction represents a combination of use-dependent and action prediction biases. Moreover, these two sources of movement bias are dissociable on the basis of movement preparation time; bias due to movement repetition is insensitive to movement

preparation time, whereas bias due to target selection is much greater when preparation time is constrained.

Finally, we considered the variability of movements made to the three probe targets, since reduced movement variance to repeated targets at the expense of bias away from alternative targets was argued by *Verstynen and Sabes (2011)* to be a signature of Bayesian adaptive tuning. Although our study was not designed to test movement variability, the dissociation that we observed for repetition versus predictive bias effects provides another source of evidence to judge whether the effects reported by *Verstynen and Sabes (2011)* reflect use-dependent or action prediction mechanisms. If their movement variability effects were dominated by use-dependent mechanisms, then movement variability should change as a function of fixed target position in our study. Contrary to this prediction, movement variance was not statistically different between trials with the fixed target at 0 and 45 (main effect of fixed target location, $F_{1, 13} = 0.73$, p=0.41), and neither were any interactions involving fixed target location. The fact that these relevant effects were not statistically significant does not provide strong evidence of no effect, but we can conclude that the data do not provide clear evidence that strictly use-dependent processes underlie history-dependent changes in movement variability.

## Possible effects of timing feedback on response execution

It is well known that the dopaminergic system responds strongly to reward and can influence response selection and vigor (*Beierholm et al., 2013*; *Niv et al., 2007*; *Bromberg-Martin et al., 2010*). Because we tried to constrain preparation time in our experiments, we provided feedback to motivate participants to adhere to the temporal constraints of the task (see Materials and methods for details). It is conceivable that any systematic variation in the nature of feedback across probe positions or preparation blocks might have influenced movement direction or vigor through processes related to reward. To examine this possibility, we analysed the percentage of trials in which participants received potentially rewarding feedback (e.g. 'good timing'). Because these percentages can only range from 0% to 100%, we used non -parametric permutation tests to analyse this type of data.

In Experiment 1, participants received the 'good timing' message on 33.5% (95%CI [28.6, 38.4]) of all trials in the long preparation block and 38.7% (95%CI [34.4, 43.1]) of trials in the short preparation block (permutation paired t-test: p=0.18, 95% CI [−0.69, 11.83]). If we consider only movements toward probe targets, a permutation analysis of variance showed that participants received more positive timing feedback in the short preparation than the long preparation block (p=0.001; Long: mean = 50%, 95% CI [41.7, 58.3]; Short: mean = 71%, 95% CI [61.8, 80.2]). More importantly, however, because the slope of the relationship between positive feedback percentage and probe positions was small, and not statistically significant (p=0.80, slope = 0.05%, 95% CI [−0.13, 0.26]), differences in the percentage of positive timing feedback received are unlikely to account for the observed effects of probe target position on movement bias and vigor.

Considering both context and probe trials in Experiment 2, any difference between long and short preparation blocks in the percentage of trials in which participants received the 'good timing' message was small with overlapping confidence intervals (Long: 31.0%, 95% CI [18.8, 43.0]; Short: and 36.98%, 95% CI [20.7, 52.9]; permutation paired t-test: p=0.18, 95% CI [−9.54, 20.85]). A permutation analysis of variance across the probe trials in both blocks revealed only a main effect of probe position (p=0.028, slope = −0.06%, 95% CI [−0.16, 0.03]), indicating participants received less positive feedback as the probe target was presented further away from the centre of the distribution. Note however that the slope of the probe position effect was small (implying ~5% difference in positive feedback trials between 90° and 0° probe targets) and that the confidence interval overlapped zero.

In experiment 3, the message 'good timing' did not vary significantly across fixed target positions (permutation anova: p=0.068). Although this effect is marginal, there was no evidence of a linear increase/decrease as fixed targets were positioned further away from probe targets (slope = −0.004%, 95% CI [−0.03, 0.02]), suggesting that the observed linear effects of fixed target position on vigor are unlikely to be due to systematic effects of timing feedback.

Overall, we did not find strong evidence for differences in timing feedback that could readily explain the core pattern of results observed for movement biases and vigor in this study. Although timing feedback effects appeared to covary with bias or vigor effects for some specific experimental

conditions, apparent associations were not consistent across experiments or preparation time conditions, and the magnitude of differences in positive feedback were small.

## Discussion

The data show that the effects of action history rely both on use-dependent processes that depend strictly on actions previously executed, and on dynamically evolving processes associated with preparation of probable future actions. In particular, directional biases toward the most likely next target direction are greater when limited time is available for movement preparation (experiments 1, 2, 4). Such sensitivity to the timing of stimulus presentation suggests that bias in these circumstances is dominated by advanced preparation of anticipated actions, rather than mere movement repetition. We nonetheless detected clear use-dependent effects in the absence of target uncertainty in experiment 3, and evidence that distinct use-dependent and action prediction effects combine to determine movement direction in experiment 4. Together, the results indicate that information obtained from action history is treated by the brain in two very different ways. In this sense, use-dependent and action prediction effects are due to separate neural processes. Our behavioural data do not allow us to identify which components of the sensorimotor control network are responsible for these putatively distinct processes. The effects could, in principle, rely on distinct populations of neurons in different brain areas, or to activity within a given brain region under distinct neural states over time (e.g. *Kaufman et al., 2014*; *Elsayed et al., 2016*).

The effects of recent movement history appear to reflect a trade-off between improved performance for commonly executed actions, at the expense of directional errors, delayed initiation and reduced vigor for alternative actions. *Verstynen and Sabes (2011)* examined such a trade-off, between reaching errors for less frequently executed actions and reduced movement variability for actions repeated more frequently. They interpreted these effects from a probabilistic perspective, motivated by recognition that uncertainty is inherent in both sensory and motor processes. Bayesian inference shows that, given this sensorimotor noise, statistically optimal behaviour takes into account both current sensory information and the probabilities with which different environmental and physical states occur (*Faisal et al., 2008*; *Harris and Wolpert, 1998*; *Najemnik and Geisler, 2005*). Moreover, because animals obtain information necessary to predict the probability of future action requirements through their previous interactions with the world, optimal behaviour should be biased in favour of past experience. Indeed, *Verstynen and Sabes (2011)* data matched the predictions of Bayesian models, suggesting recent movement history effects can approximate Bayesian inference. They also illustrated a potential biological implementation via a competitive neural network simulation that employed a Hebbian learning rule. Our current data illustrate, however, that probabilistic or neural network models of action history effects must incorporate temporal dynamics if they are to fully account for behaviour. For example, to account for our data, Bayesian models would need to be extended to allow confidence in the sensory estimate of target location to dynamically evolve following target presentation.

Accordingly, the current data fit well with various dynamic models of decision making and action selection, in which choices are simulated as the outcomes of competitive interactions between neural representations of alternative actions (e.g. *Cisek et al., 2009*; *Christopoulos et al., 2015*; *Cisek, 2006*; *Standage et al., 2011*; *Wilimzig et al., 2006*). Core to these is recognition that action selection and execution must operate dynamically in natural settings, because environmental and internal states could change at any moment. The dynamics of competitive interactions between neural representations of alternative actions have been particularly well studied in the case of saccadic eye movements, where variations in presentation timing, or the number of visual targets or distractors, have been combined with recording or microstimulation within brain regions that maintain spatial priority maps for saccadic control (e.g. *Dorris et al., 2007*; *Basso and Wurtz, 1997*; *Arcizet et al., 2011*; *Coe et al., 2002*). According to this perspective, pre-target activity in neurons associated with an anticipated action is desirable because it allows faster initiation of actions more likely to be required next. Indeed, *Dorris and Munoz (1998)* showed that more frequent movement to one of two potential saccadic targets increased pre-target activity of neurons in the superior colliculus with receptive fields including the repeated target, and that this pre-target activity corresponded to shorter saccadic reaction times.

Our current data show that limb movements are also initiated more rapidly to more probable targets, consistent with models in which reaction time is governed by an interaction between an internal urgency signal and neural activity representing action preparation (*Cisek et al., 2009*; *Dorris et al., 2007*; *Standage et al., 2011*; *Weinberg, 2016*). Alternatively, *Haith et al. (2016)* recently argued that motor initiation is independent of motor planning. In this case, in order to account for the reaction time cost that we observed for unexpected movements, motor initiation processes would have to be independently subject to history-dependent modulation based on target expectation. Future work will need to resolve this issue. More critically, our current data extend previous observations on response timing to suggest that, if movement is initiated prior to resolution of competition between potential action representations, faster reaction times toward the repeated target come at the cost of directional errors when an unexpected target is presented (see also *Marinovic et al., 2017*).

The conceptual framework of a dynamic competition between anticipated and presented targets can also account for the apparent paradox in preparation time effects evident in experiments 1 and 2. When there was minimal uncertainty about movement initiation time experiment 1 (i.e. the timed response task), bias was negligible when the target was presented 500 ms before movement initiation. In contrast, bias was substantial when the target was presented 150 ms before the GO signal in the reaction time task of experiment 2, even though a similar time of 500 ms elapsed between target presentation and movement initiation. Such effects would be expected if uncertainty about when a motor response will be required prompts greater anticipatory preparation of the expected action (e.g. *Marinovic et al., 2011*). In this case, the resolution of competition between the anticipated action and the target-directed action should take longer, leading to greater bias for a given stimulus-response duration. Alternatively, the reaction time task might affect the *gain* of competitive interactions between target-related activity and predictive activity associated with anticipated actions (*Murphy et al., 2016*). For example, *Standage et al., 2011* simulated how variations in an internally generated 'urgency' signal could modulate speed-accuracy trade-offs. Here, if urgency is high under conditions favouring speed, then each stage in the decision process occurs more rapidly but with reduced precision. Similarly, *Hanks et al., 2014* showed that the activation dynamics of neurons representing alternative saccadic response targets are contingent upon whether a perceptual discrimination task emphasises speed over accuracy (see also *Heitz and Schall, 2012*). Thus, for a given elapsed time between target presentation and response initiation, bias towards an anticipated target should increase in parallel with the urgency to respond at the time of stimulus presentation.

A particularly interesting aspect of our data is the dissociation between the temporal dynamics of movement history effects on movement direction and vigor. Both parameters were clearly affected by movement history, as illustrated by systematic dependence on the target location with respect to the repeated movement direction, but only directional bias was strongly affected by movement preparation time. This is surprising, because saccadic reaction time and vigor effects typically co-varied in previous studies, for example in response to the reward associated with targets (*Takikawa et al., 2002b*; *Itoh et al., 2003*). Such effects appear due partly to pre-target activity in the superior colliculus and basal ganglia for saccades (*Takikawa et al., 2002b*; *Ikeda and Hikosaka, 2007*; *Sato and Hikosaka, 2002*), or in cortical sensorimotor areas such as dorsal premotor cortex for arm movements (*Pastor-Bernier and Cisek, 2011*). Although it is possible that our participants found movements to the repeated targets more rewarding, because they more often led to task success (i.e. due to directional biases), any effects on movement vigor that rely on pre-target activation in neurons involved in action selection should be time-dependent, as were biases in movement direction (experiments 1 and 2, see also *Takikawa et al., 2002b*; *Itoh et al., 2003*). Thus, the effects of action history on movement vigor in the current study appear to rely on different processes from those previously identified to underlie biases in response metrics.

A candidate to explain a target-dependent vigor effect that varies little as a function of movement preparation time is the system thought to encode expected reward value, which includes the ventral pallidum (*Tachibana and Hikosaka, 2012*). The activities of some neurons in this nucleus increase upon presentation of a rewarded target, remain tonically elevated until reward delivery, and influence strongly the vigor of saccades (*Tachibana and Hikosaka, 2012*). Thus, if people develop a spatially-distributed representation of the expected value of targets according to their action history (see e.g. *Takikawa et al., 2002a*), this system would initiate a signal to modulate movement vigor *upon presentation* of frequently repeated targets. Such a signal should depend

more strongly on the target actually presented than on the pre-target state of preparation in sensori-motor areas, and therefore provides a plausible mechanism to account for our observed time-insensitive vigor effects.

## Materials and methods

### Experimental procedures

Thirty-two self-reported right-handed volunteers were tested across four experiments (seven female, age range: 18–40 years). Ten participants completed more than one experiment. All procedures were approved by the Human Medical Research Ethics Committee of the University of Queensland and written informed consent was obtained from the participants. All experiments involved an isometric wrist aiming task previously employed by our group (see *de Rugy et al., 2012a*). Participants moved a cursor from the centre of a computer monitor to peripheral targets by exerting wrist flexion-extension and ab-adduction forces. They were instructed to move the cursor as quickly and as accurately as possible through the targets. Although the task involves only very minor displacement of the limb end-point, it does require shortening of muscle fibres (and concomitant lengthening of tendons), and motion of the cursor that represents force magnitude. Thus, for simplicity of expression, we refer to the isometric actions produced in this task as 'movements' throughout the paper. The forearm was held mid-way between pronation and supination against the supports of a custom-designed rig coupled with a six degree of freedom force-torque transducer (JR3 45E15A-163-A400N60S, Woodland, CA; see *Figure 1A*). Participants had to exert either 20 N or 30 N to reach targets depending on the condition (see below). Visual stimuli were generated using Cogent 2000 graphics (available at http://www.vislab.ucl.ac.uk/cogent_2000.php) and displayed on a 19' monitor running at 60 Hz.

Experiment 1 was designed to examine the effect of preparation time on aiming bias. As depicted in *Figure 1B*, we used a *timed response paradigm* similar to that employed by Ghez and colleagues (*Ghez et al., 1989*; *Ghez et al., 1990*). This paradigm was used because it eliminates temporal uncertainty about *when* the movement should be initiated, allowing us to more effectively control the amount of preparation time in different blocks of trials. Participants (N = 10) were trained to initiate their actions in synchrony with the last of a sequence of four tones (2 Hz, 500 ms apart). Feedback was provided after every trial about the temporal error of movement initiation time with respect to the imperative tone. After trials in which temporal error was below −50 or above 50 ms, the message 'too quick' or 'too slow' was displayed on the task display. If the temporal error was within these temporal bounds, the message 'good timing' was displayed. Participants were asked to move the cursor to the visual targets as accurately as possible (i.e. slice the target with the cursor) while simultaneously matching the time constraint as closely as possible.

Trials were performed in two blocks; in the short preparation block, visual targets appeared 150 ms before the imperative tone, and in the long preparation block visual targets appeared 500 ms before the imperative tone. The order in which participants performed short and long preparation time blocks was counterbalanced. We examined the effect of repeated movements to a Gaussian distribution of 'context' targets (mean direction 45°, SD = 7.5°) upon aiming errors to occasionally presented probe targets located at 60°, 30°, 0°, −30° and −60° relative to the average of the context target distribution. Note that the context targets were randomly drawn from the Gaussian distribution and thus differed slightly between blocks and participants. By contrast, the probe targets were the same for all blocks and participants. To initially establish the statistical distribution of presented targets, each block began with 30 context trials drawn randomly from the distribution surrounding the repeated direction, followed by a pseudorandomized presentation of 110 context targets and 35 probe trials (175 trials total). Cursor position was visible throughout a trial, and 20 N was required to achieve targets in all 350 trials.

Experiment 2 tested the effect of preparation time on aiming bias in the context of a reaction time task. This meant that the required time of response initiation was more uncertain than in experiment 1 (see *Figure 1C*), and allowed us to probe whether recent movement history influences movement initiation time. In this task, participants (N = 10) had to respond as fast as possible to a single imperative tone (i.e., there were no preceding warning/anticipation tones), and feedback of the reaction time was provided after each of the 490 trials. The message 'too slow' was displayed after trials

in which the reaction time to the IS exceeded 300 ms, whereas the message 'too quick' was displayed on trials in which the reaction was shorter than 100 ms. The message 'good timing' was displayed after trials in which the reaction time fell within 100 and 300 ms. As is Experiment 1, participants were instructed to move the cursor to the targets as accurately as possible. The visual targets were presented 150 ms (short preparation) or 500 ms (long preparation) before the tone. Thus, the movement preparation time available on any trial was the sum of the stimulus-onset asynchrony between the visual target presentation and the auditory imperative, and the reaction time to the imperative. Although the same central target location of 45° was used, a broader distribution of context targets was used in this experiment (SD = 15°), and probe trials were located from −90° to 90° in relation to the average distribution of context targets (in 30° steps). Each block began with 40 context trials, followed by a pseudo-randomised sequence of 49 probe trials interleaved with 156 context trials. As for experiment 1, the precise positions of context targets differed between blocks but the probe targets were identically placed. Cursor position was visible throughout, and 20 N was required to achieve targets in all trials.

In Experiment 3, participants (N = 18) performed 264 sequences of two consecutive movements towards alternating targets (see *Figure 6A*). Because the locations of both targets in each sequence were known in advance for all trials, the task allowed us to assess aiming biases in the absence of target uncertainty. We studied aiming errors towards probe targets positioned at 22° or 90° (each in separate blocks), as a function of the direction of movements to 'fixed' targets presented at 0°, ±30°, ±60°, ±90°, ±120°, ±150° or +180° relative to each probe (see *Figure 6B*). Note that the position of the probe target was consistent within each block, that all trials to each fixed target were performed consecutively, and that participants were explicitly informed about these task features. Thus, the positions of both probe and fixed targets were known to the participants at all times except when there was a transition from a run of one fixed targets to the next fixed target. The three trials following a transition in fixed target position were not analysed (see below).

The timed response protocol (as in Experiment 1) was used to encourage a 1 s preparation time for movements to the probe targets. Targets were displayed in synchrony with the second of a series of four tones (500 ms ISI), and the movement imperative was the fourth tone. Immediately when the cursor was returned to the origin after each movement towards a probe target, a second 'fixed target' was presented. Because we have shown previously that use-dependent biases are exacerbated by the requirement to produce large forces (*Selvanayagam et al., 2016*; *Selvanayagam et al., 2011*), fixed targets were presented further from the origin, such that the force required to reach fixed targets (30 N) was greater than that required to reach the probe targets (20 N). Participants executed 11 consecutive movement sequences involving each fixed target, and the order of fixed target presentation was random across participants. In the first three trials to each target, cursor position was visible during movements to both the probe and fixed targets, but in the next eight trials, the cursor was only visible when moving towards fixed targets. For probe targets, an expanding ring was presented to provide feedback of force magnitude but not direction. This allowed us to analyse trajectories that were unaffected by cursor feedback on previous trials to the same target. We only analysed aiming errors on probe trials without force direction feedback.

Experiment 4 was conducted to dissociate bias effects due strictly to execution of recent movements from effects due to prediction of target likelihood. Participants (N = 14) performed 320 movements to one of three probe targets (25°, 45° and 65°, 20 N magnitude), followed immediately by a movement to a fixed target (either 0° or 45° in separate blocks, 30 N magnitude, see *Figure 6C*). Thus, the first movement in each sequence of two was made to a probe target with an uncertain location, whereas there was no uncertainty regarding the location of the second fixed target. Importantly, the potential locations of the probe target on any given trial were not equally probable; the central target at 45° was presented on 60% of trials, whereas the two flanker targets at 25° and 65° were each presented on 20% of trials. When the fixed target was presented at 45°, for each probe trial the most recently executed movement (i.e. to the fixed target) was also the most likely movement to be required next (i.e. the central probe target). In contrast, when the fixed target was at 0°, the most recently executed movement (i.e. to the fixed target) was made in a different direction from the movement most likely to be required next (i.e. the central probe target). Participants were explicitly informed that the position of the fixed target was not informative about the position of the next probe target, and that these were independent events. Preparation time for movements to probe targets was controlled via the timed response protocol (150 ms or 500 ms preparation times),

such that there were four conditions; long and short preparation time trials with the fixed target at both 0° and 45°. Each of the four conditions involved 80 sequences of two consecutive movements. The first 40 sequences were context trials in which full cursor feedback was provided for movements to both the probe and fixed targets. There followed 40 trials in which only force magnitude feedback was provided via an expanding ring during probe trials. Again, we only analysed aiming errors on probe trials without force direction feedback.

No explicit power analyses were conducted to determine sample sizes, however, for Experiments 1 and 2 we used a similar sample size (N = 10) to that employed by *Verstynen and Sabes (2011)* (N = 8). In our experiments 1 and 2, we obtained large effect sizes (Experiment 1: Partial $\eta^2$ = 0.81; Experiment 2: Partial $\eta^2$ = 0.28). We wanted to reduce the chance that we would fail to detect (potentially) relatively smaller effects due to repetition alone in Experiments 3 and 4, so we aimed for a larger sample than that obtained in Experiments 1 and 2. The final sample sizes were determined by our capacity to recruit participants in a continuous period; we stopped each experiment and analysed that data when it became difficult to find new volunteers. Effect sizes for Experiments 3 and 4 were also large (Experiment 3: Partial $\eta^2$ = 0.25; Experiment 4: Partial $\eta^2$ = 0.68). Post-hoc analysis showed that for our primary measure power ranged from 0.72 to 0.99 across all experiments.

## Data reduction and analysis

Wrist forces were recorded at 2000 Hz using a National Instruments PCI data acquisition card (BNC 2090A). Data reduction was performed using custom Matlab software (Mathworks). Forces exerted along *x* and *y* axes were transformed to two-dimensional screen coordinates (e.g. cursor position) and filtered using a low-pass second order Butterworth filter with a cut-off frequency of 10 Hz. Movement onsets were estimated from the tangential speed time series (derived by numerical differentiation of the filtered cursor position data) via the algorithm recommended by *Teasdale et al. (1993)*. Movement direction was computed as the angle between the initial position of the cursor at movement onset, and its position 100 ms later. This timing is similar to that used by *Verstynen and Sabes (2011)* and reflects the feedforward phase of the movement (*Elliott et al., 2001*), before feedback mechanisms can affect cursor trajectory (*Desmurget and Grafton, 2000*). Directional error was defined as the difference between movement direction and the direction of the target. Preparation time was defined as the time between target appearance and the time of movement onset. Response vigor was defined as the peak rate of change in force achieved on each trial.

For statistical analysis, we took within-subject medians of directional error, preparation time and peak rate of force development for each probe position and timing condition. Statistical tests were performed using R (*R Core Team, 2016*). The analyses of variance were conducted using the function ezAnova (ez package). Linear trends were performed using the lm function (stats package). Bootstrapped 95% confidence intervals were obtained using the functions boot and boot.ci (boot package). Plots were generated using the ggplot function (ggplot2 package). All error bars correspond to the within-participants standard error of the mean (*Morey, 2008*). Trials in which participants moved before target presentation were discarded. Approximately 1% of all probe trials were discarded based on this criterion or because participants failed to move before the end of the trial. These three dependent variables were submitted to separate repeated measures analysis of variance for each experiment. The data were subjected to Mauchley's test of sphericity, and corrections were made to the degrees of freedom where necessary using Huynh-Feldt's method. The degrees of freedom presented in the results section are corrected. The 95% confidence intervals (CI) for pairwise differences and slopes were calculated using a bootstrap resampling method with 2000 iterations. Post hoc linear and quadratic trend analyses were used to assess how movement preparation time affected aiming bias. For Experiments 1 and 2, cumulative distribution functions (CDF) were computed for each individual's directional error and movement vigor scores across all probe targets, and then averaged across the group. The CDFs were ordered according to preparation time quantiles, allowing us to average vigor and bias values across subjects according to the initiation time of each movement, from longest to shortest preparation times. For example, the bias value for each individual at the fifth percentile for preparation time is a weighted average of aiming biases from trials with the 14th and 13th shortest preparation times (i.e. the fifth earliest movement initiation time assuming 100 trials; actual values per condition obtained by linear interpolation within the full set of 14 trials per target and condition).

## Acknowledgements

We thank Timothy Welsh for comments on the manuscript, and Hesam Alavi for assistance with data collection. WM was supported by the Australian Research Council - DE120100653. TC was supported by the Australian Research Council - FT120100391. The experiments were realised using Cogent 2000, developed by the Cogent 2000 team at the FIL and the ICN, and Cogent Graphics developed by John Romaya at the LON at the Wellcome Department of Imaging Neuroscience. The authors declare no competing financial interests.

## Additional information

### Funding

| Funder | Grant reference number | Author |
| --- | --- | --- |
| Australian Research Council | DE120100653 | Welber Marinovic |
| Australian Research Council | FT120100391 | Timothy J Carroll |

The authors declare that the Australian Research Council had no role in study design, data collection and interpretation, or the decision to submit the work for publication.

### Author contributions

Welber Marinovic, Conceptualization, Data curation, Formal analysis, Investigation, Methodology, Writing—original draft, Writing—review and editing; Eugene Poh, Conceptualization, Formal analysis, Investigation, Methodology, Writing—original draft; Aymar de Rugy, Conceptualization, Methodology, Writing—original draft, Writing—review and editing; Timothy J Carroll, Conceptualization, Formal analysis, Methodology, Writing—original draft, Writing—review and editing

### Author ORCIDs

Welber Marinovic http://orcid.org/0000-0002-2472-7955
Eugene Poh https://orcid.org/0000-0003-1719-000X
Aymar de Rugy http://orcid.org/0000-0001-5645-3680
Timothy J Carroll http://orcid.org/0000-0003-0761-1819

### Ethics

Human subjects: All procedures were approved by the Human Medical Research Ethics Committee of the University of Queensland and written informed consent was obtained from the participants.

### Decision letter and Author response

Decision letter https://doi.org/10.7554/eLife.26713.017
Author response https://doi.org/10.7554/eLife.26713.018

## Additional files

### Supplementary files

• Transparent reporting form
DOI: https://doi.org/10.7554/eLife.26713.016

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
