## [Decision Letter]

Thank you for submitting your article "Action history influences subsequent movement via two distinct processes" for consideration by *eLife*. Your article has been reviewed by two peer reviewers, and the evaluation has been overseen by a Reviewing Editor and Sabine Kastner as the Senior Editor. The reviewers have opted to remain anonymous.

The reviewers have discussed the reviews with one another and the Reviewing Editor has drafted this decision to help you prepare a revised submission.

Summary:

Marinovic et al., propose existence of two separable effects of movements history on movement direction; use-dependent plasticity and probability estimation. In a series of experiments, they manage to disassociate these two effects as "temporally-stable processes that are strictly use-dependent" and "dynamically-evolving and context-dependent processes that reflect prediction of future actions". The paper is generally well written, the experiments are quite clear, with a number of the clear effects. However, each of the three reviewers have some concerns to be shared with the authors. These concerns are summarized herein.

Essential revisions:

1) There is some concern related to the nature of the "separated processes".

A) The authors central ideas are expressed in the Introduction "Movement history provided the contextual information (is) necessary to predict the probability of future action requirements in past experiments". However, it is unclear to what extent movement direction biases are due to use-dependent processes that depend strictly on movement repetition, or due to history-dependent predictions of future action requirements. Finally, they state that "If both factors contribute, it is unknown how they interact, or are co-represented in the brain." The hidden hypothesis is that if processes are indeed separate, the dissociated behavioral processes have different representations in the brain (see also examples in Results section, subsection “Experiment 3 – Bias varies as a function of angle from a repeated action in the absence of target uncertainty”, Discussion section). In the Discussion section the authors conclude that "the results imply that use-dependent and action prediction effects are due to separate neural processes, which could rely on distinct populations of neurons […]"

This could be a strong conclusion, but the authors admit also that it could actually be one process and one population that creates the dissociated result ("[…] or to activity within a given brain region under distinct neural states over time” (Discussion section). Thus, the interpretations, abstract and discussion over-state the dissociation of the two processes they well-tested, and under-state the notion that the "strict use-dependent" is relatively a small (distinct) effect, which strongly interact with other aspects of motor control. The results could imply that the resulting behavior is controlled by one process with temporal dynamics, with a gradual shift of weights of various parameters (like Bayesian distributions. With temporal weights of past movements and sensory inputs, context, previous skills, and learned internal models that predicts the future sensory inputs and required actions.

B) The question of temporal aspects and the dynamics of the biases effect is discussed in the Introduction and Discussion but was not properly addressed in the analysis of the results. Even if we do not fully accept the argument of our comment above, it is potentially interesting, because the two effects bias may have different temporal properties (if they are represented separately or not). The extreme case will be that the strictly use-dependent (history effect) is affected only by the recent trial and does not reflect a summation over history events.

2) Relation between the different experiments (1–4) and the explanations of the different results: While the argument is clear, and some of the results indeed support the existence of separable (behaviorally observed phenomena, other experimental results are not entirely consistent with the main hypothesis. These inconsistencies should be explicitly addressed; Experiments 1 and 2 show, using different experimental protocols that biases towards the more frequent, movement direction are seen only when 'reparation time' is short. Biases are not seen when targets are shown to the subjects 500ms before movement initiation.

In experiments 3 and 4 the authors introduce a markedly different protocol of two consecutive movements – the first is the probe movement (that has a certain probability) and the second is a fixed target. This protocol allows differentiating between the effect of target probability (that is specific to the probe and the fix targets) and overall movement history (that should be the combination of both). Experiment 3 shows that when probe target is shown 1000ms before movement onset there is still a bias in hand direction which is related to the fixed target. This experiment therefore, suggests that the bias is affected by history and not just by probability (assuming that subjects are aware of the structure of that perturbation). The reviewer suggests that the existence of biases with some tuning properties in Experiment 3, where targets were indicated 1000ms before movement onset, is not consistent with the lack of apparent biases in Experiments 1 and 2. This should be better explained.

Experiment 4 introduces a manipulation of preparation time in the context of the two consecutive movements to demonstrate that the history-related bias is not affected by preparation time, whereas the probability related bias does. Here, bias in the long preparation condition is not consistent with lack of bias in Experiments 1 and 2. Additionally, the tuning which is seen in Experiment 3 of the history effect is not seen when examining the same kind of biases in Experiment 4. In Figure 2 there is a trend in the short condition that should be addressed and explained more clearly.

3) Materials and methods: The description of experiments is not sufficient for understanding the manipulation and designs (let alone for reproducing the results). For example: were the same probes presented in the same block in all experiments? A simple description of the order of events in each experiment will greatly help to follow the different designs.

Analysis: what were the measures that were used for each of the statistical tests? These measures should be clearly stated in results with the presentation of the statistical results.

Feedback and instructions: What were the accuracy instructions for the subjects in terms of movement initiation in the timed response task? Were they supposed to reduce their movement initiation feedback to 0? Feedback difference about timing may provide an alternative explanation for some of the results. For example, subjects were much more accurate in movement onsets in the short condition compared to the long in Experiment 1. Were subjects rewarded for successful timing? Same for Experiment 2 – were subjects instructed to be spatially accurate? Were they given any information about spatial accuracy? What was the temporal accuracy structure of feedback?

Another crucial point is whether subjects notified about the structure of target presentation in Experiments 3 and 4, and specifically – were they told that the fixed targets are not indicative about the upcoming probe targets?

How exactly movement onset was defined? Did the author apply a threshold?

4). Everywhere report estimates of crucial effects (such as linear trends in bias) and report their CIs, instead of only reporting F- and p- values.

5) Whenever you claim lack of effect, base it on effect estimates ("small"/"negligible" effect) and not exclusively on non-significant p>0.05.

6) Figure 3 and Figure 5 could potentially be improved by showing how bias depends on preparation time, and not on "deciles".

7) Explain the difference in estimates of use-dependent bias between experiments 3 and 4 (~5° vs. ~10°). (And also, the differences in slope for the short-prep group between Experiment 1 and Experiment 2: ~1/2 vs. ~15/90).

[Editors' note: further revisions were requested prior to acceptance, as described below.]

Thank you for resubmitting your work entitled "Action history influences subsequent movement via two distinct processes" for further consideration at eLife*eLife*. Your revised article has been favorably evaluated by Sabine Kastner (Senior editor), a Reviewing editor, and two2 reviewers.

The manuscript has been improved but there are some remaining issues that need to be addressed before acceptance, as outlined below.

Reviewer #1 (General assessment and major comments (Required)):

I think the authors addressed all the comments well, and the presentation is now clear and convincing.

Reviewer #2 (General assessment and major comments (Required)):

The authors have addressed my concerns. Given the possible role of motivation and reward on vigor and biases, I find the analysis of feedback differences between the conditions important and suggest that it will be added to the manuscript.

---

## [Author Response]

Essential revisions:1) There is some concern related to the nature of the "separated processes".A) The authors central ideas are expressed in the Introduction "Movement history provided the contextual information (is) necessary to predict the probability of future action requirements in past experiments". However, it is unclear to what extent movement direction biases are due to use-dependent processes that depend strictly on movement repetition, or due to history-dependent predictions of future action requirements. Finally, they state that "If both factors contribute, it is unknown how they interact, or are co-represented in the brain." The hidden hypothesis is that if processes are indeed separate, the dissociated behavioral processes have different representations in the brain (see also examples in Results section, subsection “Experiment 3 – Bias varies as a function of angle from a repeated action in the absence of target uncertainty”, Discussion section). In the Discussion section the authors conclude that "the results imply that use-dependent and action prediction effects are due to separate neural processes, which could rely on distinct populations of neurons […]"This could be a strong conclusion, but the authors admit also that it could actually be one process and one population that creates the dissociated result ("[…] or to activity within a given brain region under distinct neural states over time” (Discussion section). Thus, the interpretations, abstract and discussion over-state the dissociation of the two processes they well-tested, and under-state the notion that the "strict use-dependent" is relatively a small (distinct) effect, which strongly interact with other aspects of motor control. The results could imply that the resulting behavior is controlled by one process with temporal dynamics, with a gradual shift of weights of various parameters (like Bayesian distributions. With temporal weights of past movements and sensory inputs, context, previous skills, and learned internal models that predicts the future sensory inputs and required actions.

We mean to make the strong conclusion that there are really two distinct neural processes, but acknowledge that this depends on what one means by the term “process”. We think it is clear that there are both strictly use-dependent and predictive components to the bias effects, even if these components are considered part of the overall “process” of motor preparation, broadly defined. This implies that information obtained from movement history is being treated (i.e. processed) in two very different ways by the brain. We think that uncertainty about whether or not a single neural population could mediate both of our behaviourally observed components is something of a red herring. The behavioural effects must be implemented within the sensorimotor control network, which involves multiple processing nodes and stages. Clearly, we cannot identify which specific components of the network underpin our behavioural observations. The statement we made in the original discussion referring to Kaufman’s et al., (2014) paper was intended to acknowledge that two distinct processes could in principle occur in the same motor area (e.g. PMd or M1), but at different stages of motor preparation – as defined by distinct epochs separated by a state transition.

We agree with the reviewer that our effects could conceivably be generated within a single integrative network that receives multiple inputs (sensory, context, etc) with temporally evolving weights. However, in order to account for our data with a single population model, the spatial and temporal tuning of either the sensory/context inputs or the weighting vectors would have to differ markedly for strictly use-dependent versus predictive influences. Thus, the information obtained from movement history is being treated (i.e. processed) in two very different ways by the brain. Whether this differential processing is occurring in one or many nodes in the sensorimotor control network seems beside the general point. From this perspective, we consider the two bias components that we observed to reflect distinct neural processes. More generally, if you consider components that contribute to preparation to be “distinct processes” if they have different emergent properties (i.e. temporal dynamics and spatial tuning), and can combine additively – then we think that our data provide clear evidence of two processes. We provide an abbreviated coverage of this issue in the revised manuscript, and now define what we mean when we conclude that history effects operate by two distinct processes, as follows (Discussion section):

“Together, the results indicate that information obtained from action history is treated by the brain in two very different ways. In this sense, use-dependent and action prediction effects are due to separate neural processes. Our behavioural data do not allow us to identify which components of the sensorimotor control network are responsible for these putatively distinct processes. The effects could, in principle, rely on distinct populations of neurons in different brain areas, or to activity within a given brain region under distinct neural states over time (e.g. Kaufman et al., 2014, Elsayed et al., 2016).”

B) The question of temporal aspects and the dynamics of the biases effect is discussed in the Introduction and Discussion but was not properly addressed in the analysis of the results. Even if we do not fully accept the argument of our comment above, it is potentially interesting, because the two effects bias may have different temporal properties (if they are represented separately or not). The extreme case will be that the strictly use-dependent (history effect) is affected only by the recent trial and does not reflect a summation over history events.

This is an interesting point. Our experiments were designed to provide information about temporal dynamics within a trial – to probe how bias evolves as the time available to process target information increases. However, we agree that the temporal dynamics with which history dependent effects develop over multiple trials is an important issue for future studies to consider. As the reviewer points out, it is possible that the two distinct processes that we identified develop with different dynamics. We think it is unlikely that the strictly use-dependent bias is solely determined by the most recent trial, as bias in involuntary responses to non-invasive brain stimulation accumulates over time. Moreover, although our experiments were not designed to address this issue, we ran some simple comparisons on data from Experiment 3; between the mean bias from the first two movements made for each fixed target with those from the last two movements (subsection “Experiment 3 – Bias varies as a function of angle from a repeated action in the absence of target uncertainty”). Remember that in this experiment participants performed all movements towards each fixed target in a serial block, thus providing an opportune situation to analyse cumulative effects of movement history. The results suggest that strictly use-dependent bias does accumulate during a short block of trials. Note that in the process of conducting this analysis, we were forced to consider targets at the same angle from the probe target (i.e.45° clockwise and 45° counter-clockwise fixed targets) separately (we calculated individual subject medians, and then averaged between clockwise and anticlockwise targets to obtain the bias angle for analysis of tuning), whereas in the original manuscript we took the median bias of the pooled movements to all fixed targets at a given absolute angle. This change resulted in small differences in group effect sizes, but did not change the overall pattern of results.

We also added a sentence to highlight this issue (subsection “Experiment 3 – Bias varies as a function of angle from a repeated action in the absence of target uncertainty”): “An important issue that was not the specific focus of the current study is the temporal dynamics according to which bias effects accumulate over multiple trials.”

2) Relation between the different experiments (1–4) and the explanations of the different results: While the argument is clear, and some of the results indeed support the existence of separable (behaviorally observed phenomena, other experimental results are not entirely consistent with the main hypothesis. These inconsistencies should be explicitly addressed; Experiments 1 and 2 show, using different experimental protocols that biases towards the more frequent, movement direction are seen only when 'reparation time' is short. Biases are not seen when targets are shown to the subjects 500ms before movement initiation.

We first make the general point that all of the experiments have different task features that we expect to influence the two putative history dependent processes in different ways. As such, we have no expectation that effect sizes should be of similar magnitude in the different experiments, and avoid direct contrasts of effect size unless the comparisons imply particularly clear conclusions. In the case of the apparent absence of a strictly use dependent effect in experiments 1 and 2, we think there are two important issues. First is the fact that we provided full visual feedback of force trajectories in these experiments, but not in experiments 3 and 4. Thus, errors due to bias were observable and may have been corrected through error-based learning. Secondly, trajectories were more variable in general with the broader distribution of potential targets in experiments 1 and 2, which may have reduced the ability to detect small effects. On this point, we note that the grand mean bias effects are greater than zero despite the lack of statistically significant linear trends. In sum, we expect that a strictly use-dependent effect was present in these experiments but not detected (i.e. not statistically significant) due to the specific features of the study (either masked by error-based learning or small with respect to behavioural noise). Because we assumed that strictly use-dependent effects do, in general, occur we used different designs in experiments 3 and 4 to expose the putative effect. We have added the following sentences to the manuscript to make this rationale explicit (subsection “Experiment 3 – Bias varies as a function of angle from a repeated action in the absence of target uncertainty”).

“Critically, we also removed visual feedback of movements made to probe targets. We suspected that a failure to detect substantial bias effects due to strictly use-dependent processes in the first two experiments occurred because movement errors due to bias were observable and therefore may have been corrected. Thus, error-based learning may have masked strictly use-dependent bias effects in these circumstances. We therefore anticipated that removing visual feedback during assessment of bias should provide the optimal conditions to study the properties of use-dependent bias.”

In experiments 3 and 4 the authors introduce a markedly different protocol of two consecutive movements – the first is the probe movement (that has a certain probability) and the second is a fixed target. This protocol allows differentiating between the effect of target probability (that is specific to the probe and the fix targets) and overall movement history (that should be the combination of both). Experiment 3 shows that when probe target is shown 1000ms before movement onset there is still a bias in hand direction which is related to the fixed target. This experiment therefore, suggests that the bias is affected by history and not just by probability (assuming that subjects are aware of the structure of that perturbation). The reviewer suggests that the existence of biases with some tuning properties in Experiment 3, where targets were indicated 1000ms before movement onset, is not consistent with the lack of apparent biases in experiments 1 and 2. This should be better explained.

We think that our previous response addresses this point, but we have added the following section to the results of Experiment 3 to clarify the issue in the manuscript (subsection “Experiment 3 – Bias varies as a function of angle from a repeated action in the absence of target uncertainty”).

“When comparing bias effects between experiments, it appears that the “pure” repetition-dependent bias identified in Experiment 3 is weaker (i.e. <7º vs >15º) and more local than the time-sensitive effects exposed experiments 1 and 2. Even more strikingly, there is an apparent absence of strictly use-dependent bias effects in experiments 1 and 2, despite clear evidence of such in Experiment 3. This may relate to the fact that full visual feedback of movement trajectories was available to subjects in the first two experiments. We speculate that the processes that cause use-dependent biases are a general consequence of repeated action, but that the behavioural expression of such biases can be masked by error-based learning.”

Experiment 4 introduces a manipulation of preparation time in the context of the two consecutive movements to demonstrate that the history-related bias is not affected by preparation time, whereas the probability related bias does. Here, bias in the long preparation condition is not consistent with lack of bias in experiments 1 and 2. Additionally, the tuning which is seen in Experiment 3 of the history effect is not seen when examining the same kind of biases in Experiment 4. In Figure 2 there is a trend in the short condition that should be addressed and explained more clearly.

Again, our previous responses address the general point that we do not expect identical results for the different experiments due to differences in task characteristics. We acknowledge that the spatial tuning pattern that is clear in experiment 3 is not obvious in experiment 4, but highlight two points. First, since there are only four force targets in experiment 4, we expect that strictly use-dependent effects should be generated with respect to each target (although presumably more strongly for the fixed target that is repeated every second trial and with greater vigour). This would likely complicate the spatial pattern of bias observed in this experiment. Secondly, the angles between the fixed target at 0° and the three probe targets (at 25, 45 & 65°) are close to the plateau region of the tuning function revealed in Experiment 3. Thus, we are not surprised that the concave spatial tuning effect that is clear in experiment 3 is much less apparent in experiment 4.

Please see our response on statistical treatment below for our general philosophy regarding the discussion of non-significant trends. In the particular case of Figure 2, whether or not there is a real effect of delayed movement initiation with probe target eccentricity is not critical to our overall interpretations. Any tendency to greater preparation time with more peripheral targets should reduce the size of our bias effects, and the issue of whether response initiation is affected by history is addressed in experiment 2 – which was specifically designed to address this issue and where the trend is statistically significant (Figure 4). We think the paper is already long and dense, and prefer to focus on the critical issues that are most relevant to the overall conclusions of the paper, rather than provide exhaustive analysis of all marginal results.

3) Materials and methods: The description of experiments is not sufficient for understanding the manipulation and designs (let alone for reproducing the results). For example: were the same probes presented in the same block in all experiments? A simple description of the order of events in each experiment will greatly help to follow the different designs.

We have added additional description of methods in the Results section and Materials and methods section.

Analysis: what were the measures that were used for each of the statistical tests? These measures should be clearly stated in results with the presentation of the statistical results.

We have now explained each measure prior to presentation of results in each Results section.

Feedback and instructions: What were the accuracy instructions for the subjects in terms of movement initiation in the timed response task? Were they supposed to reduce their movement initiation feedback to 0? Feedback difference about timing may provide an alternative explanation for some of the results. For example, subjects were much more accurate in movement onsets in the short condition compared to the long in Experiment 1. Were subjects rewarded for successful timing? Same for Experiment 2 – were subjects instructed to be spatially accurate? Were they given any information about spatial accuracy? What was the temporal accuracy structure of feedback?

We apologise for the oversight in failing to adequately explain this in the original manuscript. In experiment 1, participants were asked to initiate their actions in synchrony with the last of a sequence of 4 tones, as per training (see Materials and methods section), and told to move the cursor to the visual targets as accurately as possible (e.g. slice the target with the cursor) (see Materials and methods section). If they succeeded to initiate their actions within a temporal window of +/- 50 ms in relation to the IS, a feedback message "good timing" was displayed on the monitor after trial completion. Feedback about temporal error in Experiment 1 had 3 levels: too early (<-50 ms in relation to IS), too late (>50 ms in relation to IS), and good timing (>-50 and <50 ms in relation to the IS). This feedback structure is now explained in the methods section of the revised manuscript (see Materials and methods section). As similar feedback structure was used for Experiment 2 but the reaction time window for the message “good timing” was > 100 and < 300 ms in relation to the IS (see Materials and methods section). No external feedback on aiming error was provided, but subjects could see their full cursor trajectory with respect to the target.

The reviewer suggests that differences in timing accuracy in long and short preparation blocks might explain some of the results. For example, if participants were more temporally accurate in the short block, they would receive more positive reinforcement in the short preparation block than in the long preparation block. To examine whether participants received a higher percentage of "good timing" feedback in one of the two blocks, we analysed the percentage of trials participants received a rewarding feedback (e.g. good timing). In Experiment 1, the average percentage of good timing feedback across all trials was 33.5% (95%CI [28.6, 38.4]) and 38.7% (95%CI [34.4, 43.1]) in the long and short blocks, respectively. Because these percentages can only range from 0 to 100%, we used non -parametric permutation tests to analyse this type of data. A permutation paired t-test failed to indicate a statistically significant difference between the means in long and short preparation blocks (P = 0.18, 95%CI [-0.69, 11.83]). Thus, any difference in positive feedback between blocks was small. Similarly, when considering both probe and context trials from experiment 2 there were only small differences between means in the long (mean = 31%, 95%CI [18.8, 43.0]) and short (mean = 36.9%, 95%CI [20.7, 52.9]) preparation blocks, P = 0.18, difference 95% CI [-9.54, 20.85]. However, if we consider only probe trials, participants did receive more rewarding feedback in the short than in the long preparation block (P = 0.001) in experiment 1, but we failed to observe an effect of probe position that could explain our results (P = 0.80; Slope = 0.06, 95%CI [-0.13, 0.26]). For experiment 2, analysis of probe trials showed no effect of preparation time block, but a significant main effect of probe position, such that more positive feedback was obtained for probe targets closer to the centre of the context target distribution. However, the magnitude of the slope of this effect was small (slope = -0.06, implying ~5% difference in positive feedback between 0 and 90° probe targets – ie on average the greater positive feedback at 0° than at 90° was less than one trial out of the available 14) and the bootstrapped interval confidence wide (95%CI [-0.16, 0.03]). In Experiment 3, we found a statistically significant linear effect on response vigour (slope = -0.13, 95%CI [-0.22, -0.05]) despite the fact that any trend for rewarding feedback (“good timing”) to decrease as a function of fixed target distance was negligible (slope = -0.004, 95%CI [-0.03, 0.02]).

It seems to us that these subtle differences in feedback timing are unlikely to account for our key bias and vigor effects across experiments. We wrote the following section for possible inclusion in the manuscript at the end of the Results section, but prefer to omit it. The paper is already long, and we fear that the central message of the paper will be more difficult for the reader to appreciate if we include more data. If the reviewers and editors prefer its inclusion as a condition of publication, we can obviously do so.

Possible effects of timing feedback on response execution

"It is well known that the dopaminergic system responds strongly to reward and can influence response selection and vigor (Beierholm et al., 2013, Niv et al., 2007, Bromberg-Martin et al., 2010). Because we tried to constrain preparation time in our experiments, we provided feedback to motivate participants to adhere to the temporal constraints of the task (see Materials and methods section for details). It is conceivable that any systematic variation in the nature of feedback across probe positions or preparation blocks might have influenced movement direction or vigor through processes related to reward. To examine this possibility, we analysed the percentage of trials in which participants received potentially rewarding feedback (e.g. “good timing”).

In Experiment 1, participants received the "good timing" message on 33.5% (95%CI [28.6, 38.4]) of all trials in the long preparation block and 38.7% (95%CI [34.4, 43.1]) of trials in the short preparation block (permutation paired t-test: P = 0.18, 95%CI [-0.69, 11.83]). If we consider only movements toward probe targets, a permutation analysis of variance showed that participants received more positive timing feedback in the short preparation than the long preparation block (p=0.001; Long: mean = 50%, 95% CI [41.7, 58.3]; Short: mean = 71%, 95%CI [61.8, 80.2]). More importantly, however, because the slope of the relationship between positive feedback percentage and probe positions was small, and not statistically significant (p = 0.80, slope = 0.05%, 95%CI [-0.13, 0.26]), differences in the percentage of positive timing feedback received are unlikely to account for the observed effects of probe target position on movement bias and vigor.

Considering both context and probe trials in Experiment 2, any difference between long and short preparation blocks in the percentage of trials in which participants received the "good timing" message was small with overlapping confidence intervals (Long: 31.0%, 95%CI [18.8, 43.0]; Short: and 36.98%, 95%CI [20.7, 52.9]; permutation paired t-test: P = 0.18, 95%CI [-9.54, 20.85]). A permutation analysis of variance across the probe trials in both blocks revealed only a main effect of probe position (p=0.028, slope = -0.06%, 95% CI [-0.16, 0.03]), indicating participants received less positive feedback as the probe target was presented further away from the centre of the distribution. Note however that the slope of the probe position effect was small (implying ~5% difference in positive feedback trials between 90° and 0° probe targets) and that the confidence interval overlapped zero.

In Experiment 3, the message "good timing" did not vary significantly across fixed target positions (permutation anova: P = 0.068). Although this effect is marginal, there was no evidence of a linear increase/decrease as fixed targets were positioned further away from probe targets (slope = -0.004%, 95%CI [-0.03, 0.02]), suggesting that the observed linear effects of fixed target position on vigor are unlikely to be due to systematic effects of timing feedback.

Overall, we did not find strong evidence for differences in timing feedback that could readily explain the core pattern of results observed for movement biases and vigor in this study. Although timing feedback effects appeared to covary with bias or vigor effects for some specific experimental conditions, apparent associations were not consistent across experiments or preparation time conditions, and the magnitude of differences in positive feedback were small."

Another crucial point is whether subjects notified about the structure of target presentation in Experiments 3 and 4, and specifically – were they told that the fixed targets are not indicative about the upcoming probe targets?

They were explicitly notified in both cases. In Experiment 3, participants were informed that the probe target was either 22° or 90° for all trials within a block, and that targets would alternate exclusively between this target and a series of alternative targets (presented in blocks). We have made additions to the text to make this point clearer (see Materials and methods section). In Experiment 4, participants were told explicitly that the first target would be placed in one of three positions randomly and that the subsequent movement would be towards 45° (centre of the distribution of the targets) or 0°. We have added text to emphasize this point to the reader (see Materials and methods section). Thus, for both experiments, subjects knew that the fixed targets were not indicative of upcoming probe targets.

How exactly movement onset was defined? Did the author apply a threshold?

In the manuscript we stated that movement onset time was calculated using the derivative of the tangential force time-series according to the algorithm recommended by Teasdale et al., (1993). This information is provided in subsection “Data reduction and Analysis” of the revised manuscript. In more detail, this algorithm first locates the sample at which the force derivative exceeds 10% of its maximum value (Vmax). Then it traces back from this point and stops at the first sample (S) less than or equal to V_max_/10 – V_max_/100. Next the algorithm determines the standard deviation of the series between sample 1 and sample S (SD). Working back from S, onset is the first sample less than or equal to S-SD.

4). Everywhere report estimates of crucial effects (such as linear trends in bias) and report their CIs, instead of only reporting F- and p- values.

Following the reviewer’s suggestion, we now provide descriptive statistics of important effects and also the bootstrapped confidence intervals for linear trend slopes (2000 iterations) to support our interpretations.

5) Whenever you claim lack of effect, base it on effect estimates ("small"/"negligible" effect) and not exclusively on non-significant p>0.05.

We appreciate the thrust of the reviewer’s comment, and have revised all of our text to avoid the impression that we interpret a lack of statistically significant effects as strong evidence that no effect is truly present. As indicated above, we now report additional estimates of critical effect sizes and 95% confidence intervals. However, it would be unworkable to provide detailed description of every possible effect that we test in the paper. Our general approach is to conduct omnibus anovas to identify effects that are sufficiently large and consistent to meet conventional thresholds of statistical significance. Where multi-level main or interaction effects are present, we then conduct post-hoc linear (&/or quadratic) trend analyses. The Results section is long and dense, and we are reluctant to expand it further by exhaustive quantitative description of effects when the omnibus anova does not reveal any main effects or interactions. The reader can see the effect sizes qualitatively in the graphs (and quantify them using the provided source data), and draw their own conclusions about what non-significant effects might mean.

6) Figure 3 and Figure 5 could potentially be improved by showing how bias depends on preparation time, and not on "deciles".

We think that using deciles is the best way to illustrate the effect. Please see our response to the detailed query on this issue below.

7) Explain the difference in estimates of use-dependent bias between Experiments 3 and 4 (~5° vs. ~10°). (And also, the differences in slope for the short-prep group between Experiment 1 and Experiment 2: ~1/2 vs. ~15/90).

We have made additional statements that explain the apparent discrepancies in effect size between the different experiments in the relevant Results sections. The difference in slope between Experiments 1 and 2 seems almost certain to be due to the dramatic difference in preparation time available to process the target information, and to differences in the width of the context target distributions. We already address these differences in preparation time between experiments at some length in the results and discussion, but have added additional sentences that are specific to the reviewer’s point (see subsection “Experiment 2 – Bias depends on the interaction between preparation time and the urgency to move”):

“Note that these biases in movement direction are much smaller than those observed in Experiment 1. […] Finally, probe target locations were uncertain in Experiment 4, but not in Experiment 3, which might have tended to exacerbate bias effects in Experiment 4.

[Editors' note: further revisions were requested prior to acceptance, as described below.]

[…] Reviewer #2:The authors have addressed my concerns. Given the possible role of motivation and reward on vigor and biases, I find the analysis of feedback differences between the conditions important and suggest that it will be added to the manuscript.

This analysis has been added to Results section of the revised manuscript.